# GenRL: Multimodal-foundation world models for generalization in embodied agents

**Pietro Mazzaglia***
IDLab, Ghent University

**Tim Verbelen**
VERSES AI Research Lab

**Bart Dhoedt**
IDLab, Ghent University

**Aaron Courville**
Mila, University of Montreal

**Sai Rajeswar**
ServiceNow Research

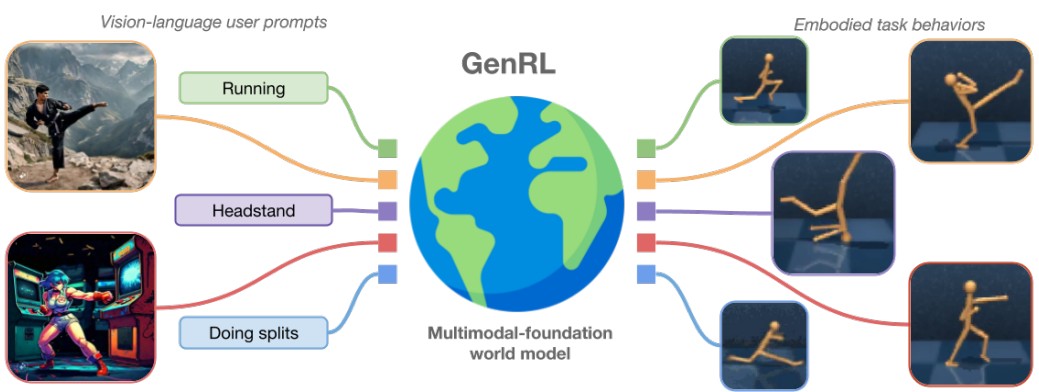

Figure 1: *Multimodal-foundation world models* connect and align the video-language space of a foundation model with the latent space of a generative world model for reinforcement learning, requiring vision-only data. Our *GenRL* framework turns visual and/or language prompts into latent targets and learns to realize the corresponding behaviors by training in the world model's imagination.

## Abstract

Learning generalist embodied agents, able to solve multitudes of tasks in different domains is a long-standing problem. Reinforcement learning (RL) is hard to scale up as it requires a complex reward design for each task. In contrast, language can specify tasks in a more natural way. Current foundation vision-language models (VLMs) generally require fine-tuning or other adaptations to be adopted in embodied contexts, due to the significant domain gap. However, the lack of multimodal data in such domains represents an obstacle to developing foundation models for embodied applications. In this work, we overcome these problems by presenting multimodal-foundation world models, able to connect and align the representation of foundation VLMs with the latent space of generative world models for RL, without any language annotations. The resulting agent learning framework, GenRL, allows one to specify tasks through vision and/or language prompts, ground them in the embodied domain's dynamics, and learn the corresponding behaviors in imagination. As assessed through large-scale multi-task benchmarking in locomotion and manipulation domains, GenRL enables multi-task generalization from language and visual prompts. Furthermore, by introducing a data-free policy learning strategy, our approach lays the groundwork for foundational policy learning using generative world models.

**Website, code and data:** `mazpie.github.io/genrl`

---

*Work done while interning at Mila/ServiceNow Research. Email: `pietro.mazzaglia@ugent.be`

38th Conference on Neural Information Processing Systems (NeurIPS 2024).

# 1 Introduction

Foundation models are large pre-trained models endowed with extensive knowledge of the world, which can be readily adapted for a given task [44]. These models have demonstrated extraordinary generalization capabilities in a wide range of vision [28, 45, 67] and language tasks [43, 19, 53, 11]. As we aim to extend this paradigm to embodied applications, where agents physically interact with objects and other agents in their environment, we require generalist agents that are capable of reasoning about these interactions and executing action sequences within these settings [61].

Reinforcement learning (RL) allows agents to learn complex behaviors from visual and/or proprioceptive inputs [18, 26, 27] by maximizing a specified reward function. Scaling up RL to multiple tasks and embodied environments remains challenging as designing reward functions is a complicated process, requiring expert knowledge and prone to errors which can lead to undesired behaviors [1]. Recent work has proposed the adoption of visual-language models (VLMs) to specify rewards for visual environments using language [4, 48, 39], e.g. using the similarity score computed by CLIP [45] between an agent's input images and text prompts. However, these approaches mostly require fine-tuning of the VLM [38], otherwise, they tend to work reliably only in a few visual settings [48].

In most RL settings, we lack multimodal data to train or fine-tune domain-specific foundation models, due to the costs of labelling agents' interactions and/or due to the intrinsic unsuitability of some embodied contexts to be converted into language. For instance, in robotics, it's non-trivial to convert a language description of a task to the agent's actions which are hardware-level controls, such as motor currents or joint torques. These difficulties make it hard to scale current techniques to large-scale generalization settings, leaving open the question:

*How does one effectively leverage foundation models for generalization in embodied domains?*

In this work, we present GenRL, a novel approach requiring no language annotations that allows training agents to solve multiple tasks from visual or language prompts. GenRL learns multimodal-foundation world models (MFWMs), where the joint embedding space of a foundation video-language model [57] is connected and aligned with the representation of a generative world model for RL [23], using only vision data. The MFWM allows the specification of tasks by grounding language or visual prompts into the embodied domain's dynamics. Then, we introduce an RL objective that enables learning to accomplish the specified tasks in imagination [24], by matching the prompts in latent space.

Compared to previous work in world models and VLMs for RL, one emergent property of GenRL is the possibility to generalize to new tasks in a completely data-free manner. After training the MFWM, it possesses both strong priors over the dynamics of the environment, and large-scale multimodal knowledge. This combination enables the agent to interpret a large variety of task specifications and learn the corresponding behaviors. Thus, analogously to foundation models for vision and language, GenRL allows generalization to new tasks without additional data and lays the groundwork for foundation models in embodied RL domains [44].

# 2 Preliminaries and background

Additional related works can be found in Appendix A.

**Problem setting.** The agent receives from the environment observations $x \in \mathcal{X}$ and interacts with it through actions $a \in \mathcal{A}$. In this work, we focus on visual reinforcement learning, so observations are images of the environment. The objective of the agent is to accomplish a certain task $\tau$, which can be specified either in the observation space $x_\tau$, e.g. through images or videos, or in language space $y_\tau$, where $\mathcal{Y}$ represents the space of all possible sentences. Crucially, compared to a standard RL setting, we do not assume that a reward signal is available to solve the task. When a reward function exists, it is instead used to evaluate the agent's performance.

**Generative world models for RL.** In model-based RL, the optimization of the agent's actions is done efficiently, by rolling out and scoring imaginary trajectories using a (learned) model of the environment's dynamics. In recent years, this paradigm has grown successful thanks to the adoption of generative world models, which learn latent dynamics by self-predicting the agent's inputs [23]. World models have shown impressive performance in vision-based environments [24],

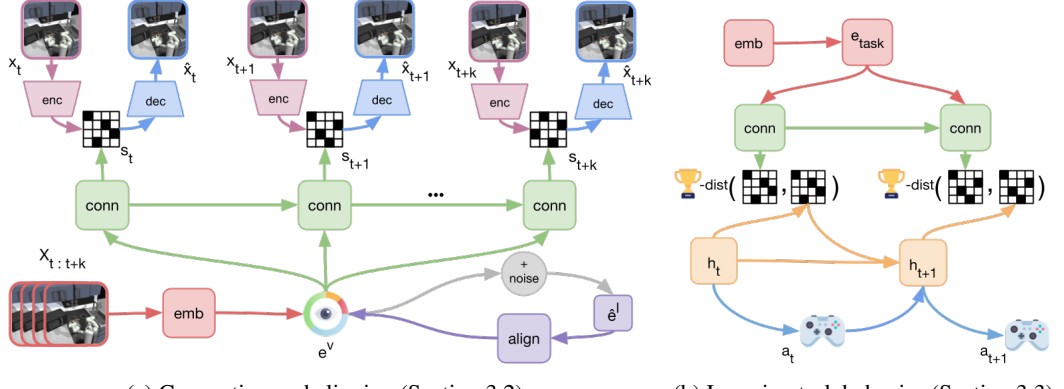

(a) Connecting and aligning (Section 3.2)  (b) Learning task behavior (Section 3.3)

Figure 2: *Overview of GenRL.* The agent learns a multimodal-foundation world model that connects and aligns (a) the representation of a foundation VLM with the latent states of a generative world model. Given a certain task prompt, (b) the model allows embedding the task and translating into targets in the latent dynamics space, which the agent can learn to achieve by using RL in imagination.

improving our ability to solve complex and open-ended tasks [26]. Generative world models have been successfully extended to many applications, such as exploration [51], skill learning [42], solving long-term memory tasks [50], and robotics [58, 16].

**Foundations models for RL.** Large language models (LLMs) have been used for specifying behaviors using language [41, 29, 56, 59], but this generally assumes the availability of a textual interface with the environment or that observations and/or actions can be translated to the language domain. The adoption of vision-language models (VLMs) reduces these assumptions, as it allows the evaluation of behaviors in the visual space. However, this approach has yet to show robust performance, as it generally requires fine-tuning of the VLM [4, 15], prompt hacking techniques [9] or visual modifications to the environment [4].

**Vision-language generative modelling.** Given the large success of image-language generative models [49], recent efforts in the community have focused on replicating and extending such success to the video domain, where the temporal dimension introduces new challenges, such as temporal consistency and increased computational costs [30, 3]. Video generative models are similar to world models for RL, with the difference that generation models outputs are typically not conditioned on actions, but rather conditioned on language [30] or on nothing at all (i.e. an unconditional model).

## 3 GenRL

### 3.1 World models for RL

GenRL learns a task-agnostic world model representation by modelling the sequential dynamics of the environment in a compact discrete latent space $S$ [24, 26]. Latent states $s \in S$ are sampled from independent categorical distributions. The gradients for training the model are propagated through the sampling process with straight-through estimation [5].

The world model is made of the following components:

| | | | |
|---|---|---|---|
| Encoder: | $q_\phi(s_t|x_t),$ | Sequence model: | $h_t = f_\phi(s_{t-1}, a_{t-1}, h_{t-1}),$ |
| Decoder: | $p_\phi(x_t|s_t),$ | Dynamics predictor: | $p_\phi(s_t|h_t),$ |

trained with the loss:

$$\mathcal{L}_\phi = \sum_t \underbrace{D_{\mathrm{KL}}\big[q_\phi(s_t|x_t)\|p_\phi(s_t|s_{t-1}, a_{t-1})\big]}_{\text{dyn loss}} - \underbrace{\mathbb{E}_{q_\phi(s_t|x_t)}\big[\log p_\phi(x_t|s_t)\big]}_{\text{recon loss}}, \quad (1)$$

where $p_\phi(s_t|s_{t-1}, a_{t-1})$ is a shorthand for $p_\phi(s_t|f_\phi(s_{t-1}, a_{t-1}, h_{t-1}))$. The sequence model is implemented as a linear GRU cell [8]. Differently from recurrent state space models (RSSM; [25]),

for our framework, encoder and decoder models are not conditioned on the information present in the sequence model. This ensures that the latent states only contain information about a single observation, while temporal information is stored in the hidden state of the sequence model. Given the simpler encoder-decoder strategy of our model, the encoder can be seen as a probabilistic visual tokenizer, which is grounded in the target embodied environment [64].

## 3.2 Multimodal-foundation world models

Multimodal VLMs are large pre-trained models that have the following components:

$$\text{Vision embedder:} \quad e^{(v)} = f_{\text{PT}}^{(v)}(x_{t:t+k}), \qquad \text{Language embedder:} \quad e^{(l)} = f_{\text{PT}}^{(l)}(y),$$

where $x_{t:t+k}$ is a sequence of visual observations and $y$ is a text prompt. For video-language models, $k$ is generally a constant number of frames (e.g. $k \in \{4, 8, 16\}$ frames). Image-language models are a special case where $k = 1$ as the vision embedder takes a single frame as an input. For our implementation, we adopt the InternVideo2 video-language model [57] (with $k$=8).

To connect the representation of the multimodal foundation VLM with the world model latent space, we instantiate two modules: a *latent connector* and a *representation aligner*:

$$\text{Connector:} \quad p_\psi(s_{t:t+k}|e), \qquad \mathcal{L}_{\text{conn}} = \sum_t D_{\text{KL}}\big[p_\psi(s_t|s_{t-1}, e)\|\text{sg}(q_\phi(s_t|x_t))\big],$$

$$\text{Aligner:} \quad e^{(v)} = f_\psi(e^{(l)}), \qquad\qquad \mathcal{L}_{\text{align}} = \|e^{(v)} - f_\psi(e^{(l)})\|_2^2,$$

where $\text{sg}(\cdot)$ indicates to stop gradients propagating.

The connector learns to predict the latent states of the world model from embeddings in the VLM's representation space. The connector's objective consists of minimizing the KL divergence between its predictions and the world model's encoder distribution. While more expressive architectures, such as transformers [55] or state-space models [21] could be adopted, we opt for a simpler GRU-based architecture for video modelling. This way, we keep the method simple and the architecture of the connector is symmetric with respect to the world model's components.

**Aligning multimodal representations.** The connector learns to map visual embeddings from the pretrained VLM to latent states of the world model. When learning the connector from visual embeddings $e^{(v)}$, we assume it can generalize to the (theoretical) corresponding language embedding $e^{(l)}$ if the angle $\theta$ between the two embeddings is small enough, as shown in Fig. 3a. This can be expressed as $\cos\theta > c$ or $\theta < \arccos c$, with $c$ a small positive constant [68].

Multimodal VLMs trained with contrastive learning exhibit a *multimodality gap* [34], where the spherical embeddings of different modalities are not aligned. Given a dataset of vision-language data, this projective function can be learned. However, in embodied domains vision-language data is typically unavailable. Thus, we have to find a way to align the representations using *no language annotations*.

Previous methods inject the noise into the vision embeddings during training [66, 68]. This leads to the situation shown in Figure 3b, where $c$ grows larger with the noise. This allows language embeddings to be close enough to their visual counterparts.

In our work, we instead learn an aligner network, which maps points surrounding $e^{(v)}$ closer to $e^{(v)}$. As represented in Fig. 3c, this way, $c$ is unaltered but the aligner will map $e^{(l)}$ close enough to $e^{(v)}$. Since we use noise to sample points around $e^{(v)}$ the aligner model can be trained using vision-only data and thus, no language annotations.

Figure 3: When training the connector on (a) the VLM's representation we can address the multimodality gap in multiple ways: (b) prior works adopt noise during the training of the connector, (c) we adopt an aligner network that learns to map points in proximity of the visual embedding close the corresponding embedding.

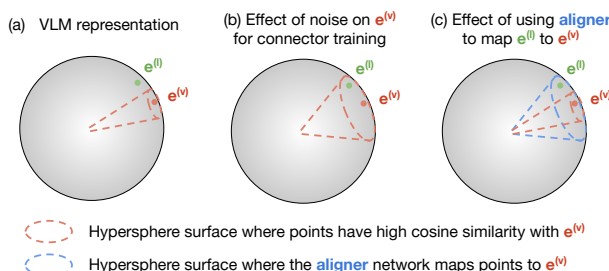

The aligner allows us to train a noise-free connector, which has two main advantages: (i) it yields higher prediction accuracy for visual embedding inputs while maintaining a similar alignment for language embedding inputs; and (ii) it is more flexible; it's easier to re-train/adapt for different noise levels, as it only requires re-training the aligner module, and its use can be avoided if unnecessary.

### 3.3 Specifying and learning tasks in imagination

World models can be used to imagine trajectories in latent space, using the sequential and dynamics models. This allows us to train behavior policies in a model-based RL fashion [24]. Given a task specified through a visual or language prompt, our MFWM can generate the corresponding latent states by turning the embedder's output, $e_{\text{task}}$, into sequences of latent states $s_{t:t+k}$ (decoded examples are shown in Figure 1). The objective of the policy model $\pi_\theta$ is then to match the goals specified by the user by performing trajectory matching.

The trajectory matching problem can be solved as a divergence minimization problem [13], between the distribution of the states visited by the policy $\pi_\theta$ and the trajectory generated using the aligner-connector networks from the user-specified prompt:

$$\theta = \arg\min_\theta \mathbb{E}_{a_t \sim \pi_\theta(s_t)} \left[ \sum_t \gamma^t \texttt{distance}\big(p_\phi(s_{t+1}|s_t, a_t) \| p_\psi(s_{t+1}|e_{\text{task}})\big) \right], \quad \text{with} \quad e_{\text{task}} = f_{\text{PT}}(\cdot).$$

(2)

The KL divergence is a natural candidate for the distance function [13]. However, in practice, we found that using the cosine distance between linear projections of the latent states notably speeds up learning and enhances stability. We can then turn the objective in Eq. 2 into a reward for RL:

$$r_{\text{GenRL}} = \cos\big(g_\phi(s_{t+1}^{\text{dyn}}), g_\phi(s_{t+1}^{\text{task}})\big), \quad \text{with} \quad s_{t+1}^{\text{dyn}} \sim p_\phi(s_{t+1}|s_t, a_t), s_{t+1}^{\text{task}} \sim p_\psi(s_{t+1}|e_{\text{task}}), \quad (3)$$

where $g_\phi$ represents the first linear layer of the world model's decoder. We train an actor-critic model to maximize this reward and achieve the tasks specified by the user [26]. Additional implementation details are provided in Appendix B.

**Temporal alignment.** One issue with trajectory matching is that it assumes that the distribution of states visited by the agent starts from the same state as the target distribution. However, the initial state generated by the connector may differ from the initial state where the policy is currently in. For example, consider the Stickman agent on the right side of Figure 1. If the agent is lying on the ground and tasked to run, the number of steps to get up and reach running states may surpass the temporal span recognized by the VLM (e.g. in our case 8 frames), causing disalignment in the reward.

To address this initial condition alignment issue, we propose a *best matching trajectory* technique, inspired by best path decoding in speech recognition [20]. Our technique involves two steps:

1. We compare the first $b$ states of the target trajectory with $b$ states obtained from the trajectories imagined by the agent by sliding along the time axis. This allows one to find at which timestep $t_a$ the trajectories are best aligned (the comparison provides the highest reward).

2. We align the temporal sequences in the two possible contexts: (a) if a state from the agent sequence comes before $t_a$, the reward uses the target sequence's initial state; and (b) if the state comes $k$ steps after $t_a$, it's compared to the $s_{t+k}$ state from the target sequence.

In all experiments, we fix $b = 8$ (number of frames of the VLM we use [57]), which we found to strike a good compromise between comparing only the initial state ($b = 1$) and performing no alignment ($b = $ imagination horizon). An ablation study can be found in Appendix E.

## 4 Experiments

Overall, we employ a set of 4 locomotion environments (Walker, Cheetah, Quadruped, and a newly introduced Stickman environment) [54] and one manipulation environment (Kitchen) [22], for a total of 35 tasks where the agent is trained without rewards, using only visual or language prompts. Details about the datasets, tasks, and prompts used can be found in the Appendix C.

Table 1: *Language-to-action in-distribution.* Offline RL from language prompts on tasks that are included in the agent's training dataset. Scores are episodic rewards averaged over 10 seeds (± standard error) rescaled using min-max scaling with $(\text{min} = \text{random policy}, \text{max} = \text{expert policy})$.

| | Image-language VLM | | | | Video-language VLM | | | | |
|---|---|---|---|---|---|---|---|---|---|
| | IQL | TD3+BC | TD3 | WM-CLIP | IQL | TD3+BC | TD3 | WM-CLIP | GenRL |
| walker stand | 0.67 ± 0.03 | 0.92 ± 0.02 | 0.93 ± 0.03 | 1.01 ± 0.0 | 0.66 ± 0.05 | 0.64 ± 0.03 | 1.01 ± 0.0 | 0.94 ± 0.01 | 1.02 ± 0.0 |
| walker run | 0.24 ± 0.03 | 0.27 ± 0.01 | 0.09 ± 0.02 | 0.05 ± 0.02 | 0.29 ± 0.02 | 0.24 ± 0.02 | 0.35 ± 0.01 | 0.7 ± 0.01 | 0.77 ± 0.02 |
| walker walk | 0.41 ± 0.05 | 0.34 ± 0.05 | 0.14 ± 0.0 | 0.21 ± 0.01 | 0.4 ± 0.03 | 0.44 ± 0.03 | 0.88 ± 0.02 | 0.91 ± 0.02 | 1.01 ± 0.0 |
| cheetah run | 0.41 ± 0.05 | 0.0 ± 0.01 | -0.01 ± 0.0 | -0.0 ± 0.0 | 0.15 ± 0.02 | -0.01 ± 0.0 | 0.37 ± 0.01 | 0.56 ± 0.03 | 0.74 ± 0.01 |
| quadruped stand | 0.56 ± 0.02 | 0.64 ± 0.04 | 0.65 ± 0.04 | 0.97 ± 0.0 | 0.52 ± 0.06 | 0.43 ± 0.05 | 0.61 ± 0.05 | 0.97 ± 0.0 | 0.97 ± 0.0 |
| quadruped run | 0.3 ± 0.03 | 0.28 ± 0.02 | 0.24 ± 0.02 | 0.27 ± 0.0 | 0.38 ± 0.03 | 0.25 ± 0.02 | 0.26 ± 0.01 | 0.61 ± 0.02 | 0.86 ± 0.02 |
| quadruped walk | 0.26 ± 0.02 | 0.31 ± 0.02 | 0.28 ± 0.01 | 0.47 ± 0.02 | 0.32 ± 0.02 | 0.28 ± 0.04 | 0.28 ± 0.02 | 0.92 ± 0.01 | 0.93 ± 0.01 |
| stickman stand | 0.45 ± 0.06 | 0.58 ± 0.04 | 0.06 ± 0.04 | 0.71 ± 0.02 | 0.43 ± 0.04 | 0.45 ± 0.05 | 0.08 ± 0.02 | 0.32 ± 0.01 | 0.7 ± 0.02 |
| stickman walk | 0.4 ± 0.04 | 0.48 ± 0.04 | 0.18 ± 0.01 | 0.23 ± 0.01 | 0.51 ± 0.02 | 0.46 ± 0.03 | 0.41 ± 0.02 | 0.65 ± 0.05 | 0.83 ± 0.01 |
| stickman run | 0.2 ± 0.01 | 0.22 ± 0.02 | 0.03 ± 0.0 | 0.19 ± 0.01 | 0.23 ± 0.02 | 0.19 ± 0.02 | 0.21 ± 0.0 | 0.35 ± 0.01 | 0.35 ± 0.01 |
| kitchen microwave | 0.06 ± 0.04 | 0.22 ± 0.11 | 0.0 ± 0.0 | 0.0 ± 0.0 | 0.01 ± 0.01 | 0.0 ± 0.0 | 0.11 ± 0.08 | 0.9 ± 0.09 | 0.97 ± 0.02 |
| kitchen light | 0.14 ± 0.04 | 0.11 ± 0.11 | 0.59 ± 0.16 | 0.1 ± 0.09 | 0.02 ± 0.01 | 0.0 ± 0.0 | 0.18 ± 0.11 | 0.26 ± 0.13 | 0.46 ± 0.09 |
| kitchen burner | 0.21 ± 0.05 | 0.18 ± 0.05 | 0.09 ± 0.05 | 0.03 ± 0.03 | 0.05 ± 0.02 | 0.02 ± 0.01 | 0.31 ± 0.1 | 0.78 ± 0.06 | 0.62 ± 0.07 |
| kitchen slide | 0.02 ± 0.01 | 0.0 ± 0.0 | 0.0 ± 0.0 | 0.0 ± 0.0 | 0.04 ± 0.02 | 0.02 ± 0.02 | 0.7 ± 0.14 | 0.88 ± 0.04 | 1.0 ± 0.0 |
| overall | 0.31 ± 0.03 | 0.33 ± 0.04 | 0.23 ± 0.04 | 0.30 ± 0.02 | 0.29 ± 0.02 | 0.24 ± 0.02 | 0.41 ± 0.05 | 0.70 ± 0.04 | 0.80 ± 0.02 |

## 4.1 Offline RL

In offline RL, the objective of the agent is to learn to extract a certain task behavior from a given fixed dataset [33]. The performance of the agent generally depends on its ability to 'retrieve' the correct behaviors in the dataset and interpolate among them. Popular techniques for offline RL include off-policy RL methods, such as TD3 [18], advantage-weighted behavior cloning, such as IQL [31], and behavior-regularized approaches, such as CQL [32] or TD3+BC [17].

We aim to assess the multi-task capabilities of different approaches for designing rewards using VLMs. We collected large datasets for each of the domains evaluated, containing a mix of structured data (i.e. the replay buffer of an agent [26] learning to perform some tasks) and unstructured data (i.e. exploration data collected using [51]). The datasets contain **no reward information** and **no text annotations** of the trajectories. The rewards for training for a given task must be inferred by the agent, i.e. using the cosine similarity between observations and the given prompt or, in the case of GenRL, using our reward formulation (Eq. 3).

We compare GenRL to two main categories of approaches:

- *Image-language rewards*: following [48], the cosine similarity between the embedding for the language prompt and the embedding for the agent's visual observation is used as a reward. For the VLM, we adopt the SigLIP-B [65] model as it's reported to have superior performance than the original CLIP [45].
- *Video-language rewards*: similar to the image-language rewards, with the difference that the vision embedding is computed from a video of the history of the last $k$ frames, as done in [15]. For the VLM, we use the InternVideo2 model [57], the same used for GenRL.

The evaluation compares GenRL to various offline RL methods from the literature, including IQL, TD3+BC, and TD3. We also introduce a model-based baseline, *WM-CLIP*. This baseline is the antithesis of GenRL as, rather than learning a connector and an aligner, it learns a "reversed connector". This module learns to predict VLM embeddings from the world model states (GenRL does the opposite). This makes it possible to compute rewards in imagination in a similar way to the model-free baselines, by computing the cosine similarity between the visual embeddings predicted from imagined states and the task's language embeddings.

All methods are trained for 500k gradient steps, and evaluated on 20 episodes. For each task, model-free agents require training the agent from scratch, including the visual encoder, actor, and critic networks on the entire dataset. Model-based agents require training the model once for each domain and then training an actor-critic for each task. Other training and baseline details are reported in Appendix D.

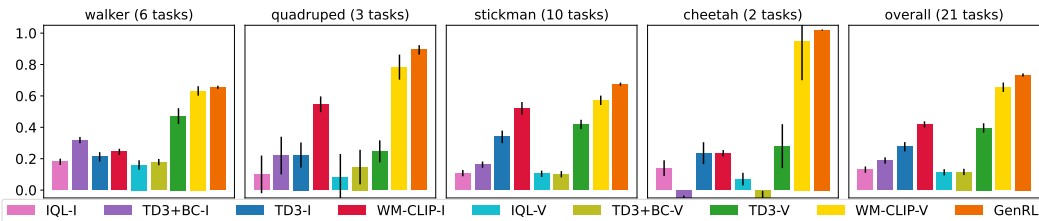

Figure 4: *Language-to-action generalization.* Offline RL from language prompts on tasks that are not deliberately included in the training dataset. Performance averaged over 10 seeds and standard error was reported with black lines. Detailed results per task in Appendix K.

**Language-to-action in-distribution.** We want to verify whether the methods can retrieve the task behaviors that are certainly present in the training data, when specifying the task only through language. We present results in Table 1, with episodic rewards rescaled so that 0 represents the performance of a random agent, while 1 represents the performance of an expert agent.

GenRL excels in overall performance across all domains and tasks, outperforming other methods particularly in dynamic tasks like walking and running in the quadruped and cheetah domains. However, in some static tasks of the kitchen domain, other methods occasionally outperform GenRL. This can be explained by the fact that the target sequences that GenRL infers from the prompt are often slightly in motion, even in static cases. To address this, we could set the target sequence length to 1 for static prompts, but we opted to maintain the method's simplicity and generality, acknowledging this as a minor limitation.

As expected, video-language rewards tend to perform better than image-language rewards for dynamic tasks. The less conservative approach, TD3, performs better than the other model-free baselines in most tasks, similarly to what is shown in [62]. The model-based baseline's performance, WM-CLIP-V, is the closest to GenRL's.

**Language-to-action generalization.** To assess multi-task generalization, we defined a set of tasks not included in the training data. Although we don't anticipate agents matching the performance of expert models, higher scores in this benchmark help gauge the generalization abilities of different methods. We averaged the performance across various tasks for each domain and summarized the findings in Figure 4, with detailed task results in Appendix K.

Overall, we observe a similar trend as for the in-distribution results. GenRL significantly outperforms all model-free approaches, especially in the quadruped and cheetah domains, where the performance is close to the specialized agents' performance. Both for image-language (-I in the Figure) and video-language (-V in the Figure) more conservative approaches, such as IQL and TD3+BC tend to perform worse. This could be associated with the fact that imitating segments of trajectories is less likely to lead to high-rewarding trajectories, as the tasks are not present in the training data.

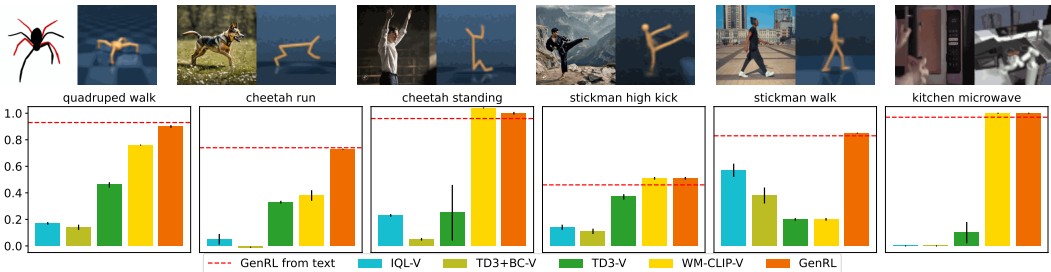

Figure 5: *Video-to-action.* GenRL allows grounding video prompts into the target environment's dynamics. It allows visualization of the model's interpretation of the prompts, using the decoder (top row), and it allows turning prompts into behaviors, leading to generally higher performance than other approaches. 10 seeds. Additional visualizations on the project website.

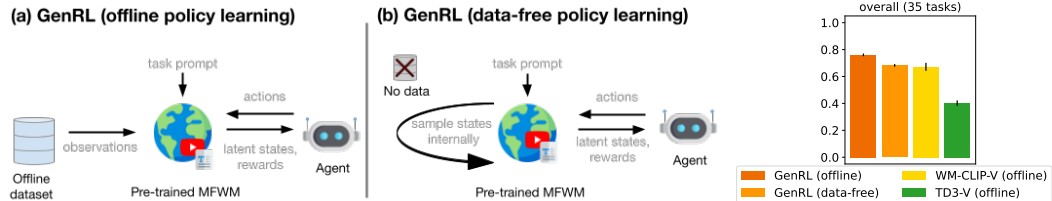

Figure 6: By removing data dependencies on actions (on-policy learning in the model's imagination), rewards (computed using only the prompt and latent states, using Eq. 3), and observations (by sampling latent states within the model), GenRL agents can be adapted for new tasks in a data-free fashion. Performance is averaged over 10 seeds and standard error is reported with black lines. Detailed results per task in Appendix K.

**Video-to-action.** While language strongly simplifies the specification of a task, in some cases providing visual examples of the task might be easier. Similarly as for language prompts, GenRL allows grounding visual prompts (short videos) into the embodied domain's dynamics and then learning of the corresponding behaviors.

In Figure 5, we provide behavior learning results from video prompts. The tasks included are of static and dynamic nature and span across 4 different domains. Visualizations of the videos used as prompts are available on the project website, where we also present a set of "grounded videos" generated by the model using the prompts (see snapshots at the top of Fig. 5). These can be obtained by inferring the latent targets corresponding to the vision prompts (left images, in the Figure) and then using the decoder model to decode reconstructed images (right images, in the Figure).

The results show a similar trend to the language prompts experiments and the performance when using video prompts is aligned to the language-to-action performance, for the same tasks. In general, we found it interesting that the VLM allows us to generalize to very different visual styles (drawings, realistic, AI-generated), very different camera viewpoints (quadruped, microwave), and different morphologies (cheetah tasks).

**Summary.** The experiments presented allow us to establish more clearly the main ingredients that contribute to the stronger performance of GenRL: (i) the video-language model helps in dynamic tasks, (ii) model-based algorithms lead to higher performance, (iii) the connection-alignment system presented generally outperforms the "reversed" way of connecting the two representations.

## 4.2 Data-free policy learning

In the previous section, we evaluated several approaches for designing reward using foundation VLMs. Clearly, model-free RL approaches require continuous access to a dataset, to train the actor-critic and generalize across new tasks. Model-based RL can learn the actor-critic in imagination. However, in previous work [26, 24], imagining sequences for learning behaviors first requires processing actual data sequences. The data is used to initialize the dynamics model, and obtain latent states that represent the starting states to rollout the policy in imagination. Furthermore, in order to learn new tasks, reward-labelled data is necessary to learn a reward model, which provides rewards to the agent during the task learning process.

Foundation models [44] are generally trained on enormous datasets in order to generalize to new tasks. The datasets used for the model pretraining are not necessary for the downstream applications, and sometimes these datasets are not even publicly available [43, 19]. In this section, we aim to establish a new paradigm for foundation models in RL, which follows the same principle of foundation models for vision and language. We call this paradigm *data-free policy learning* and we define it as the ability to generalize to new tasks, after pre-training, by learning a policy completely in imagination, with no access to data (not even to the pre-training dataset).

GenRL enables data-free policy learning thanks to two main reasons: the agent learns a task-agnostic MFWM on a large varied dataset during pre-training, and the MFWM enables the possibility of specifying tasks directly in latent space, without requiring any data. Thus, in order to learn behaviors in imagination, the agent can: `(i)` sample random latent states in the world model's representation,

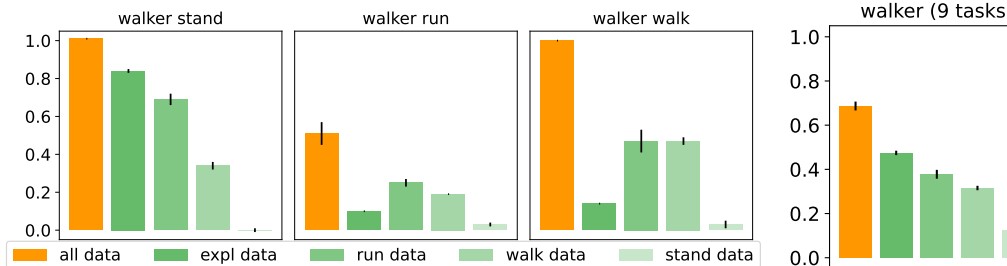

Figure 7: *Training data distribution*. Analysing the impact of the training data distribution on the generalization performance of GenRL. Performance is obtained by training behaviors in data-free mode, after training the MFWM on different subsets of the training dataset. Performance averaged over 10 seeds (black lines indicate standard error). Full results in Appendix K.

(ii) rollout sequences in imagination, following the policy's actions, and (iii) compute rewards, using the targets obtained by processing the given prompts with the connector-aligner networks.

In Figure 6, we provide a diagram that further clarifies the differences between training GenRL in an offline and in a data-free policy learning fashion. Then, we present results that compare data-free policy learning with offline RL baselines, as discussed in Section 4.1. While data-free policy learning shows a slight decrease in overall performance, its performance remains close to the original GenRL's performance and still outperforms other approaches. In Appendix K, we further show that the difference in performance is minimal across most domains, and data-free policy learning even performs better in the kitchen domain.

By employing data-free learning, after pre-training, agents can master new tasks without data. By requiring no CPU-GPU memory transfers of the data, data-free policy learning also reduces the training time of the policy, often allowing convergence within only 30 minutes of training. As we scale up foundation models for behavior learning, the ability to learn data-free will become crucial. Although very large datasets will be employed to train future foundation models, GenRL adapts well without direct access to original data, offering flexibility where data may be proprietary, licensed or unavailable.

## 4.3 Analysis of the training data distribution

As demonstrated in Sections 4.1 and 4.2, after training on a large dataset, a GenRL agent can adapt to multiple new tasks without additional data. The nature of the training data, detailed in Appendix C, combines exploration and task-specific data. Thus, we ask ourselves what subsets of the data are the most important ones for GenRL's training.

To identify critical data types for GenRL, we trained different MFWMs on various dataset subsets. Then, we employ data-free behavior learning to train task behaviors for all tasks. We present an analysis over subsets of the walker dataset in Figure 7.

The results confirm that a diverse data distribution is crucial for task success, with the best performance achieved by using the complete dataset, followed by the varied exploration data. Task-specific data effectiveness depends on task complexity, for instance, 'run data' proves more useful and generalizable than 'walk data' or 'stand data' across tasks. Crucially, 'stand data', which shows minimal variation, limits learning for a general agent but can still manage simpler tasks like 'lying down' and 'sitting on knees' as detailed in Appendix K.

Moving forward with training foundation models in RL, it will be essential to develop methods that extract multiple behaviors from unstructured data and accurately handle complex behaviors from large datasets. Thus, the ability of GenRL to primarily leverage unstructured data is a significant advantage for scalability.

# 5  Discussion

We introduced GenRL, a world-model based approach for grounding vision-language prompts into embodied domains and learning the corresponding behaviors in imagination. The multimodal-foundation world models of GenRL can be trained using unimodal data, overcoming the lack of multimodal data in embodied RL domains. The data-free behavior learning capacity of GenRL lays the groundwork for foundation models in RL that can generalize to new tasks without any data.

**A framework for behavior generation.** A common challenge with using LLMs and VLMs involves the need for prompt tuning to achieve specific tasks. As GenRL relies on a foundation VLM, similar to previous approaches [4, 48] it is not immune from this issue. However, GenRL uniquely allows for the visualization of targets obtained from specific prompts. By decoding the latent targets, using the MFWM decoder, we can visualize the interpreted prompt before training the corresponding behavior. This enables a much more explainable framework, which allows fast iteration for prompt tuning, compared to previous (model-free) approaches which often require training the agent to identify which behaviors are rewarded given a certain prompt.

**Limitations.** Despite its strengths, GenRL presents some limitations, largely due to inherent weaknesses in its components. From the VLMs, GenRL inherits the issue related to the multimodality gap [34, 66] and the reliance on prompt tuning. We proposed a connection-alignment mechanism to mitigate the former. For the latter, we presented an explainable framework, which facilitates prompt tuning by allowing decoding of the latent targets corresponding to the prompts. From the world model, GenRL inherits a dependency on reconstructions, which offers advantages such as explainability but also drawbacks, such as failure modes with complex observations. We further investigate this limitation in Appendix I and present other potential limitations in Appendix J.

**Future work.** As we strive to develop foundation models for generalist embodied agents, our framework opens up numerous research opportunities. One such possibility is to learn multiple behaviors and have another module, e.g. an LLM, compose them to solve long-horizon tasks. Another promising area of research is investigating the temporal flexibility of the GenRL framework. We witnessed that for static tasks, greater temporal awareness could enhance performance. This concept could also apply to actions that extend beyond the time comprehension of the VLM. Developing general solutions to these challenges could lead to significant advancements in the framework.

## Acknowledgments and Disclosure of Funding

Pietro Mazzaglia is funded by a Ph.D. grant of the Flanders Research Foundation (FWO). This research was supported by a Mitacs Accelerate Grant.

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

# A Extended Related Work

[Linked to Section 2]

**World models.** Recent research has focused on the question of how to learn world models from large-scale video datasets [37, 60]. In [6], they leverage a latent action representation, but their work is mostly focussed on 2D platform videogames or simple robotic actions. In [14], they use frame-by-frame video prediction as a way to provide rewards for RL. DynaLang [36] studies the incorporation of language prediction as part of the world model, to train multimodal world models also from datasets without actions or rewards. The representation in DynaLang is shared in the world model between vision and language, while for GenRL, the world model representation is trained on vision-only data and connected-aligned to the multimodal foundation representation.

**Foundation models for actions.** Few cases of foundation models for embodied domains have been developed. Notable mentions are GATO [47], a large-scale behavior cloning agent, trained on 604 tasks. VPT [2] a large-scale model trained on Minecraft data, using human-expert labeled trajectories. The model learns strong behavioral priors by behavior cloning which can be fine-tuned using RL. STEVE-1 [35] connects VPT's behavioral prior with the MineCLIP model representation [15], using the unCLIP approach [46]. RT-X [12] are large-scale transformer models trained on expert robotics dataset, sharing a common action space (end-effector pose) across different embodiments.

# B Implementation details

[Linked to Section 3]

**Actor-critic.** Rewards can be maximized over time in imagination in a RL fashion, using actor-critic models of the form:

$$\text{Actor:} \quad \pi_\theta(a_t|s_t), \qquad \text{Critic:} \quad v_\theta(R_t^\lambda|s_t), \quad \text{where} \quad R_t^\lambda = r_t + \gamma[(1 - \lambda v_{t+1}) + \lambda R_{t+1}^\lambda]$$

For the actor-critic, we follow the implementation advances proposed in DreamerV3 [26] (version 1 of the paper, dated January 2023), such as using a two-hot distribution for learning the critic network and scaling returns in the actor loss.

When computing the reward $r_{\text{GenRL}}$, we use the mode of the distribution for the target $s_{t+1}^{\text{task}} \sim p_\psi(s_{t+1}|e_{\text{task}})$ to improve stability.

**Hyperparameters.** For the hyperparameters, we follow DreamerV3 [26] (version 1 of the paper, dated January 2023). Differences from the default hyperparameters or model size choices are illustrated in Table 2. For instance, a main difference is that we use difference batch sizes/lengths for training the MFWM and the actor-critic as these two stages are now independent from each other.

The connector network uses the same hyperparameters and architecture as the sequential dynamics of the world model. The aligner network employs a small U-Net, with a bottleneck that is half the size of the embedding representation. The embedding representation is 768-dimensional. Further details can be found in our accompanying code implementation.

| Name | Value |
|---|---|
| Multimodal Foundation World Model | |
| Batch size | 48 |
| Sequence length | 48 |
| GRU recurrent units | 1024 |
| CNN multiplier | 48 |
| Dense hidden units | 1024 |
| MLP layers | 4 |
| Actor-Critic | |
| Batch size | 32 |
| Sequence length | 32 |

Table 2: World model and actor-critic hyperparameters.

# C Tasks

[Linked to Section 4]

**Stickman environment.** The Stickman environment is based on the Walker environment from the *dm_control* suite. We designed the Stickman environment to explore tasks that require upper body limbs (e.g. boxing, doing a handstand) without the complexity of training a humanoid (which requires a significantly larger amount of data to be solved [63]). The number of joints is increased by 4: 2 joints per arm, one is for the shoulder, the other for the elbow. The total number of joints is 10. The action space is normalized to be in [-1,1] as all *dm_control* tasks. The robot also presents a head, to resemble a humanoid.

**Prompts and scores.** We present the list of tasks employed, along with the language prompts used for specifying the task, in Table 3. For the newly introduced tasks, the goal can be easily inferred by reading the task's name or its prompt. For the 'flipping' tasks, we consider flips both in the forward direction and backward direction, as the VLM struggles to distinguish directions. The reward functions used to evaluate the agent's score can be found in our open-source code.

The prompts we use have been fine-tuned for the InternVideo2 model [57]. However, we found that they mostly improved performance for the SigLIP model too [65]. One common observation is that these models are generally biased towards human actions. Thus, specifying the embodiment in the prompt is sometimes helpful, e.g. 'spider running fast' or 'running like a quadruped'. Another observation is that for some behaviors the agent can produce very different styles, e.g. the agent can be walking in a slow or fast way, or in a more or less composed manner. Specifying words like 'fast' or 'clean' helps clarifying what kind of behavior is expected.

Table 3: Task and prompt used for each task

| Task | Prompt | Specialized agent score | Random agent score |
|---|---|---|---|
| quadruped run | spider running fast | 930 | 10 |
| quadruped walk | spider walking fast | 960 | 10 |
| quadruped stand | spider standing | 990 | 15 |
| quadruped jump | spider jumping | 875 | 15 |
| quadruped two legs | on two legs | 875 | 14 |
| quadruped lie down | lying down | 965 | 750 |
| cheetah run | running like a quadruped | 890 | 9 |
| cheetah standing | standing like a human | 930 | 5 |
| cheetah lying down | lying down | 920 | 430 |
| stickman walk | robot walk fast clean | 960 | 35 |
| stickman run | robot run fast clean | 830 | 25 |
| stickman stand | standing | 970 | 70 |
| stickman flipping | doing flips | 790 | 45 |
| stickman one foot | stand on one foot | 865 | 20 |
| stickman high kick | stand up and kick | 920 | 55 |
| stickman lying down | lying down horizontally | 965 | 380 |
| stickman sit knees | praying | 966 | 40 |
| stickman lunge pose | lunge pose | 950 | 100 |
| stickman headstand | headstand | 955 | 180 |
| stickman boxing | punch | 920 | 80 |
| stickman hands up | standing with the hands up | 830 | 5 |
| walker walk | walk fast clean | 960 | 45 |
| walker run | run fast clean | 770 | 30 |
| walker stand | standing up straight | 970 | 150 |
| walker flipping | doing backflips | 720 | 20 |
| walker one foot | stand on one foot | 955 | 20 |
| walker high kick | stand up and kick | 960 | 25 |
| walker lying down | lying down horizontally | 975 | 170 |
| walker sit knees | praying | 945 | 100 |
| walker lunge pose | lunge pose | 945 | 150 |
| kitchen microwave | opening the microwave fully open | 1 | 0 |
| kitchen light | activate the light | 1 | 0 |
| kitchen burner | the burner becomes red | 1 | 0 |
| kitchen slide | slide cabinet above the knobs | 1 | 0 |

# D Experiments settings

[Linked to Section 4]

**Baselines.** In order to implement performant model-free offline RL baselines we adopt the findings of [52] and [7], adopting larger deeper networks and layer normalization. Inputs are 64x64x3 RGB images. We use a frame stack of 3. The encoder architecture is adapted from the DrQ-v2 encoder [63]. We did find augmentations on the images, e.g. random shifts, to hurt performance.

The WM-CLIP baseline learns a "reversed connector" from the world model representation to the VLM representation (GenRL does the opposite). The "reversed connector", given the latent state corresponding to a certain observation, predicts the corresponding embedding. Formally:

$$\text{Reversed connector:} \quad \hat{e}_t^{(v)} = f_\psi(s_t, h_t) \qquad \mathcal{L}_{\text{rev\_conn}} = \|e^{(v)} - \hat{e}_t^{(v)}\|_2^2$$

After training the reversed connector, visual embeddings can be inferred from latent states. For policy learning, rewards are computed using the cosine similarity between embeddings inferred from imagined latent states and the prompts' embedding.

The reversed connector is implemented as a 4-layer MLP, with hidden size 1024. For fair comparison, we adopt the same world model for WM-CLIP and GenRL. For WM-CLIP we pre-train the additional reversed connector, while for GenRL the connector and aligner.

**Offline RL.** For each task, training model-free agents (IQL, TD3, TD3+BC) requires re-training the full agent (visual encoder, actor, critic) on the entire dataset, from scratch, while training model-based agents (GenRL, WM-CLIP) requires training the model once for each domain and then training an actor-critic for each task. Moreover, for training the actor-critic in GenRL, we only use 50k gradient steps, as the policy converges significantly faster than for the other methods.

**Datasets composition.** We present the datasets' composition in Table 4.

Table 4: Datasets composition.

| Domain | ∼ num of observations | Subset | Subcount |
|---|---|---|---|
| walker | 2.5M | walker run | 500k |
| | | walker walk | 500k |
| | | walker stand | 500k |
| | | walker expl | 1M |
| cheetah | 1.8M | cheetah run | 1M |
| | | cheetah expl | 820k |
| quadruped | 2.5M | quadruped expl | 1M |
| | | quadruped run | 500k |
| | | quadruped stand | 500k |
| | | quadruped walk | 500k |
| kitchen | 3.6M | kitchen slide | 700k |
| | | kitchen light | 700k |
| | | kitchen bottom burner | 700k |
| | | kitchen microwave | 700k |
| | | kitchen expl | 800k |
| stickman | 2.5M | stickman stand | 500k |
| | | stickman walk | 500k |
| | | stickman expl | 1M |
| | | stickman run | 500k |
| minecraft | 4M | - | - |

**Compute resources.** We use a cluster of V100 with 16GB of VRAM for all our experiments. To enable efficient training, image and video-CLIP embeddings are computed in advance and stored with the datasets. Training the MFWM for 500k gradient steps takes ∼ 5 days. After pre-training the MFWM, training the actor-critic for a prompt for 50k gradient steps takes less than 5 hours. In data-free mode, it takes less than 3 hours. In both cases, convergence normally arrives after 10k gradient steps, but we keep training. Model-free baselines take around 7 hours to train for 500k gradient steps.

On a single GPU, model-free RL is faster to train for a small number of runs. GenRL starts becoming advantageous when using the world model for training for more than 60 runs (which is often the case, considering the number of runs = N seeds x M tasks per domain). When adopting the data-free policy learning strategy, GenRL doesn't rely on the dataset at all. This halves the time required for training, as there are no data transfers between the CPU (where the dataset is normally loaded) and the GPU for training.

# E    Temporal alignment ablation

[Linked to Section 3]

In Figure 8, we investigate the impact of our best-matching trajectory strategy for temporally aligning imaginary latent states to target states for GenRL's reward computation.

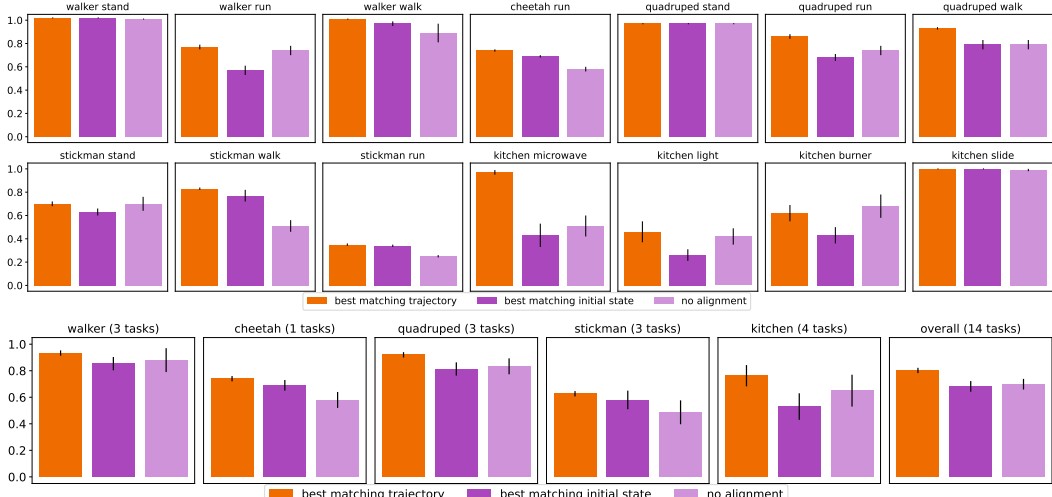

Figure 8: *Temporal alignment ablation.* We analyze the impact of temporal alignment in our proposed RL objective for matching sequential targets. Results averaged over 10 seeds.

# F    Aligner model ablation study

[Linked to Section 3]

To establish the importance of the aligner network, in Table 5 we report the results of additional ablations:

- **GenRL - no aligner:** this is an ablation of GenRL where the the language prompt's embedding is directly fed into the connector, rather than processing it first with the aligner;
- **TD3-V and WM-CLIP-V + aligner:** for these ablations, we first process the language prompt's embedding using GenRL's pre-trained aligner. Then, we use it to compute the cosine similarity for the reward function, as for the original baselines.

Table 5: Aligner ablation study.

|  | GenRL - no aligner | GenRL | WM-CLIP-V | WM-CLIP-V + aligner | TD3-V | TD3-V + aligner |
|---|---|---|---|---|---|---|
| cheetah | 0.32 ± 0.02 | 0.93 ± 0.01 | 0.82 ± 0.17 | 0.84 ± 0.02 | 0.31 ± 0.09 | 0.77 ± 0.04 |
| stickman | 0.09 ± 0.01 | 0.66 ± 0.01 | 0.54 ± 0.03 | 0.51 ± 0.03 | 0.38 ± 0.02 | 0.38 ± 0.02 |
| walker | 0.19 ± 0.01 | 0.75 ± 0.01 | 0.70 ± 0.02 | 0.74 ± 0.01 | 0.56 ± 0.04 | 0.48 ± 0.04 |
| kitchen | 0.25 ± 0.00 | 0.76 ± 0.08 | 0.71 ± 0.14 | 0.84 ± 0.09 | 0.32 ± 0.16 | 0.27 ± 0.10 |
| quadruped | 0.17 ± 0.02 | 0.91 ± 0.02 | 0.81 ± 0.04 | 0.76 ± 0.05 | 0.32 ± 0.04 | 0.33 ± 0.04 |
| overall | 0.17 ± 0.00 | 0.76 ± 0.01 | 0.67 ± 0.03 | 0.68 ± 0.02 | 0.40 ± 0.02 | 0.42 ± 0.02 |

We can observe that: i) the aligner mechanism is crucial in GenRL's functioning. ii) processing the language embedding in the reward function of the WM-CLIP-V and TD3-V baselines changes performance on some tasks (performance per domain varies). However, using the aligner provides no advantage overall.

We believe the aligner is very important in GenRL because its output, the processed language embedding, is fed to another network, the connector. If the language embeddings were not processed by the aligner, they would have been too different from the embeddings used to train the connector, which are the visual embeddings.

Instead, for the baselines, we process the language embedding with the aligner and then use it to compute a similarity score with the visual embeddings. This overall renders very similar performance to no aligner processing, hinting that the aligner network doesn't improve the cosine similarity signal. At the same time, this also suggests that the aligner network doesn't hurt the generality of the VLM's embeddings, as the cosine similarity after processing the embedding provides a similarly useful signal as before processing.

# G Comparison with LIV

[Linked to Section 4]

We also tested all baselines with the LIV's representation [40] in the Kitchen tasks. We used LIV's open-source code to download and instantiate the model. Note that the available model is the general pre-trained model, not the one fine-tuned for the Kitchen environment.

LIV's results confirm the original paper's claims (see Appendix G4) that the representation does not work well for vision-language rewards without fine-tuning on domain-specific vision-language pairs (which are unavailable in our settings, as we use no language annotations).

Table 6: Comparison with LIV [40] on kitchen tasks.

|  | IQL + LIV | TD3+BC + LIV | TD3 + LIV | WM-CLIP + LIV | GenRL |
|---|---|---|---|---|---|
| kitchen microwave | 0.03 ± 0.03 | 0.0 ± 0.0 | 0.2 ± 0.16 | 0.0 ± 0.0 | 0.97 ± 0.02 |
| kitchen light | 0.53 ± 0.24 | 0.05 ± 0.04 | 0.0 ± 0.0 | 0.0 ± 0.0 | 0.46 ± 0.09 |
| kitchen burner | 0.2 ± 0.08 | 0.0 ± 0.0 | 0.0 ± 0.0 | 0.0 ± 0.0 | 0.62 ± 0.07 |
| kitchen slide | 0.03 ± 0.03 | 0.03 ± 0.03 | 0.0 ± 0.0 | 0.67 ± 0.27 | 1.0 ± 0.0 |
| overall | 0.20 ± 0.11 | 0.02 ± 0.02 | 0.05 ± 0.07 | 0.17 ± 0.12 | 0.76 ± 0.08 |

# H Extended Discussion on Data-free Policy Learning

[Linked to Section 4]

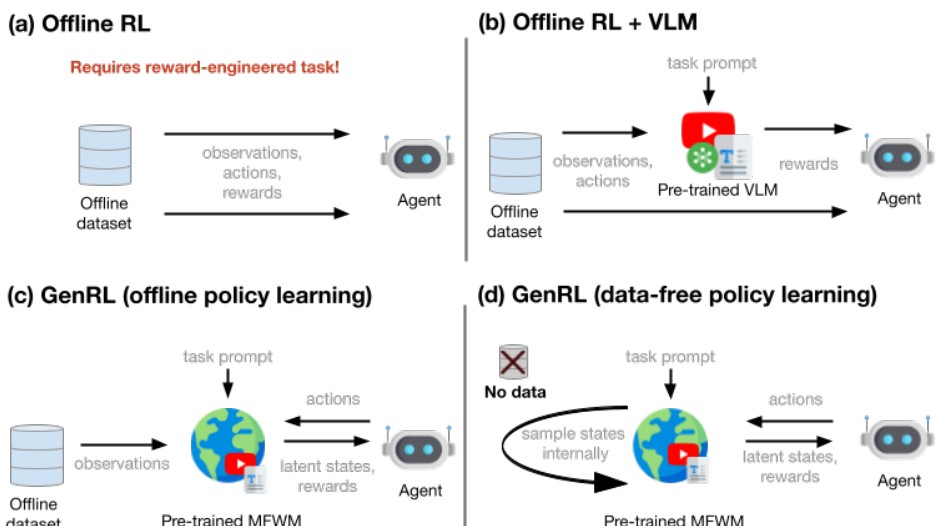

Figure 9: Representing how the different agents process the data for policy learning. By removing data dependencies on actions (on-policy learning in the model's imagination), rewards (computed using only the prompt and latent states, using Eq. 3), and observations (by sampling latent states within the model), GenRL agents can be adapted for new tasks in a data-free fashion.

Offline RL methods (Fig. 9a) learn from offline datasets of observations, actions and rewards.

Offline RL methods (Fig. 9b), combined with VLMs, can learn to perform tasks zero-shot from new prompts, but they need to sample observations and actions from the dataset for computing rewards and for policy learning.

GenRL (Fig. 9c) needs to sample (sequences of) observations from the dataset, to infer the initial latent states for learning in imagination. Afterwards, rewards can be computed on the imagined latent sequences, enabling policy learning.

Data-free GenRL (Fig. 9d), samples the initial latent states internally by combining: (i) random samples of the latent space, (ii) randomly sampled embeddings, which are mapped to "actual embeddings" using the aligner, and turned into latent states, by the connector. Thus, policy learning requires no data sampling at all.

**Initial states distribution.** Uniform sampling from the latent space of the world model often results in meaningless latent states. Additionally, the sequential dynamics model of the MFWM, using a GRU, requires some 'warmup' steps to discern dynamic environmental attributes, such as velocities.

To address these issues we perform two operations. First, we combine uniformly sampled states from the discrete latent spaces with states generated by randomly sampling the connector model, as sequences generated by the connector tend to have a more coherent structure than random uniform samples. Second, we perform a rollout of five steps using a mix of actions from the trained policy and random actions. This leads to a varied distribution of states, containing dynamic information, which we use as the initial states for the learning in imagination process.

# I  Scaling to complex observations

[Linked to Section 5]

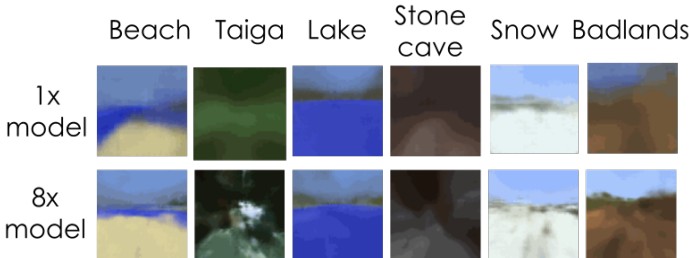

Figure 10: Decoded language prompts in Minecraft

Generalist embodied agents should be able to scale to open-ended learning settings. Using GenRL, we explored this by training an agent in the Minecraft environment using a small dataset collected by a DreamerV3 agent [26]. Note that GenRL is also trained using similar experimental settings as DreamerV3, e.g. setting up the environment in the same way. The primary challenge we found was the model's difficulty in reconstructing complex observations in this open-ended environment.

Reconstructing complex observations is a common issue with world models [10]. To overcome this limitation, while keeping the method unaltered, we attempted to scale up the number of parameters of the MFWM. Qualitative reconstruction results are presented in Figure 10. We observe that the agent is able to identify different biomes from language, even with the smaller size of the model. However, the reconstructions are significantly blurrier compared to the other environments we analyzed. When using a larger model, the reconstructions gain some details but the results still highlight the difficulty of the model in providing accurate targets from prompts.

While this might not be an issue for simple high-level tasks, e.g. 'navigate to a beach', inferring unclear targets might make it difficult to perform more precise actions, e.g. 'attack a zombie'. Future research should aim to address this issue, for instance, by improving our simple GRU-based architecture, leveraging transformers or diffusion models to improve the quality of the representation [30, 3].

# J  Additional limitations

[Linked to Section 5]

**Pre-training data requirements.** As we developed our framework, we observed that, in order to solve more complex tasks, the agent requires some expert data/demonstration of the complex behavior. We observed and analyzed this aspect in the experiments of Section 4.3. We believe this limitation is, to some extent, inevitable, as data-driven AI agents need to observe complex behaviors during training in order to be able to replicate them.

In this work, we used some exploration data, obtained using an exploration agent (Plan2Explore), and some task-specific data, collected using an expert RL agent (DreamerV3). Alternatively, one could investigate the adoption of a small set of demonstrations.

**Precise manipulation skills.** Adding additional tasks to the Kitchen environment, for testing multi-task generalization, showed to be harder than for the other domains. This is due to the difficulties of exploring meaningful behaviors in manipulation environments.

We are able to use GenRL to retrieve the four tasks present in the dataset and we are generally able to achieve similar tasks, such as "reach the microwave". From decoding the language prompts, we also see that GenRL can sensibly decode prompts such as "moving to the left" or "staying still", or more 'unusual' prompts such as

"swan pose" (where the robot arm would imitate the pose of a swan). While these prompts show that the system works well in this environment, none of these tasks would be particularly useful.

When we prompt the system with more interesting tasks that are out of the distribution, the agent generally struggles. For instance, if we ask to open the double door cabinet on the top left, the agent may reach for it but will not open it. The reason behind this is that it probably never opened that cabinet in the training dataset and thus it's impossible to reproduce such behavior in an offline manner. The same issues are most likely present with all the baselines as well, as they all use the same dataset and foundation models for training.

For the locomotion domains, instead, the exploration data is varied enough that new tasks, outside of the tasks present in the training dataset, are enabled through the exploration data/

# K    Detailed results

[Linked to Section 4]

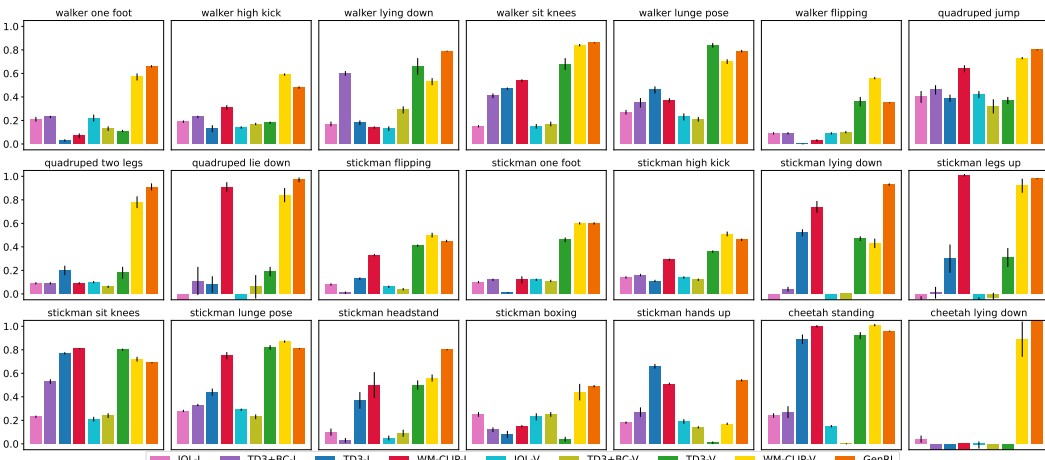

Figure 11: *Multi-task generalization detailed results.* Results averaged over 10 seeds.

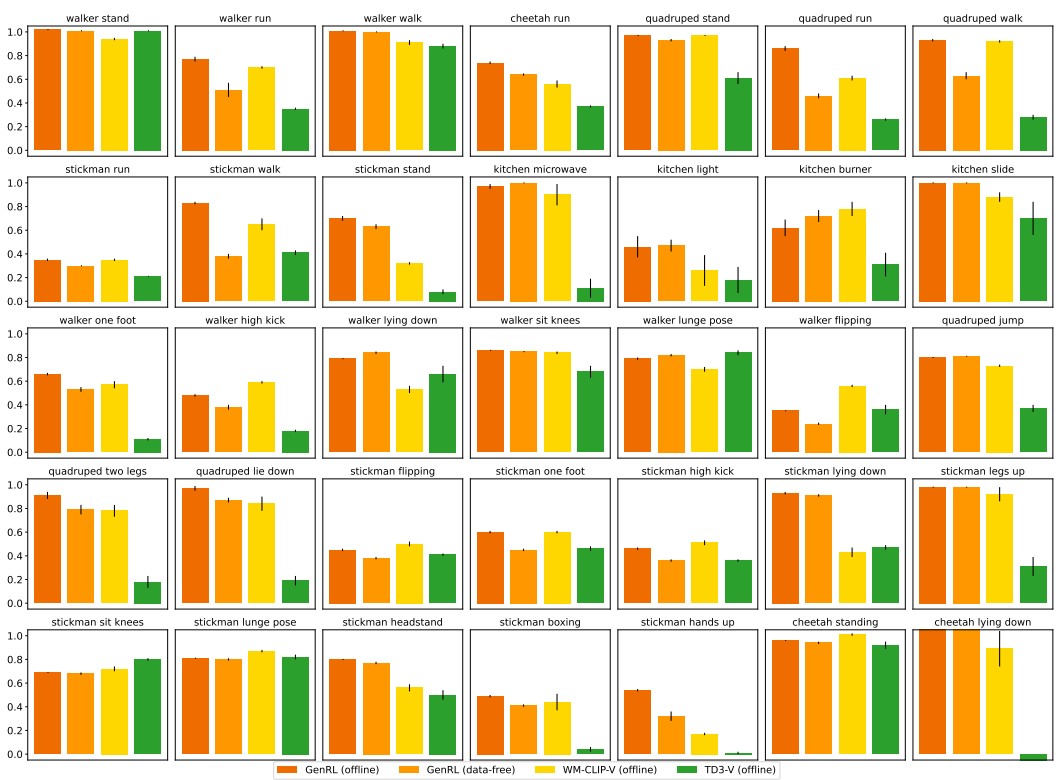

Figure 12: *Data-free RL detailed results*. Results averaged over 10 seeds.

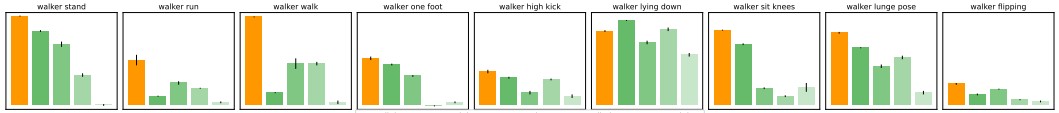

Figure 13: *Training data distribution detailed results*. Results averaged over 10 seeds.

