# OpenReview forum: "GenRL: Multimodal-foundation world models for generalization in embodied agents"
_NeurIPS.cc/2024/Conference — NeurIPS 2024 poster_

### Official Review · Reviewer_Yb6H · 2024-07-12

**Soundness:** 3
**Presentation:** 3
**Contribution:** 3
**Rating:** 5
**Confidence:** 5

**Summary:**

In this work, the authors propose learning a pixel-based reconstructive world model, and then separately learn networks to convert the representations of a pretrained VLM into the learned world model latent space.  By using a VLM trained via contrastive alignment, this essentially enables the projection of both image as well as text inputs into the latent space of the world model, and therefore simple similarity can be used to provide rewards for downstream policy learning.

**Strengths:**

This reviewer is a supporter of the idea of unifying the representation spaces of a large-scale pretrained VLM and that of a world model.  This author appreciates the benefits: matching behavior of a world model with natural language can enable text-conditioned generalization.  The preliminary experiments show promise.

**Originality**: This work appears decently original.

**Quality**: This works quality is acceptable.

**Clarity**: The clarity of the work is acceptable, the core ideas are communicated clearly.  However, there are lots of open questions surrounding this work that could be elaborated upon further.

**Significance**: This work appears to be decently significant, as a preliminary investigation in this space.

**Weaknesses:**

Chief amongst the weaknesses of this work is the limited environments applied to, and also the limited baselines (essentially, the only existing work the authors compare against is VLM-RM).  The authors can consider comparing against other forms of text-conditioned policy learning, such as LIV for the kitchen setting, or Text2Reward and similar approaches for the general case.  It also seems a strange setup to normalize in-between expert and random, and report results in this way.  This reviewer is unaware of prior work that performs evaluations in this way.  What is the rationale behind this evaluation strategy compared to what is used in prior work?

Details about certain components of the model and how they are implemented are sparse.  For example, is the aligner a video generative model (text-to-video model)?  How is it implemented?

It is a bit dissatisfying to rely on a corrupted version of vision as a language embedding.  It seems strange that the aligner should on one hand be learning to bring language embeddings meaningfully across modalities to the image/video space, which the authors motivate is necessary because of the multimodality gap.  However, the authors then treat language embeddings as a noisy corruption of a video embedding - so essentially the training objective for the aligner is essentially a denoising?  And rather than bridging a modality gap, the aligner is essentially a denoiser?

Why do we not learn the reverse direction, where we optimize a world model's latent space that projects into existing VLM space?  This design decision is not elaborated upon, but seems more intuitive to this reviewer.

From the video demonstrations, on the associated project website, it is rather unclear what is happening.  Are Behavior Retrieval videos from expert policies in an offline dataset that are matched with a particular text prompt/input video?  What are those text prompts/input videos?  It's not clear what the retrieval setup is.  For Multitask Generalization, it is also not obvious what the corresponding prompts are.  Furthermore, the results for multitask generalization do not seem smooth and natural, despite being simplistic DM_Control environments (especially the case for their proposed simplified Stickman environment) and they are missing Kitchen environments.  In the end, it appears that their method is still good as a retrieval technique (retrieving already-achieved expert behaviors in "Behavior retrieval") due to the underlying VLM, and is decent at reconstructing video prompts, but still suffers in terms of learning coherent policies (e.g. what is visualized in "Multitask generalization"), which is ultimately what is of interest.

For the video prompts that are decoded, it appears as if almost all of them are rather stationary (with the exception of the cheetah/dog example and the human dancing example) - they collapse to a stationary goal pose.  Perhaps this is because the clips are so short (8 frames) that essentially it boils down to pose-matching.  It is not obvious that this is that beneficial in supervising motion; so why does this improve upon static image supervision?  Indeed, many of the results that are shown learned by the policy are rather stationary and do not have much movement (most are just jitters around a stationary pose).  It then begs the question how this approach improves upon just a static goal supervision.  However, the authors simultaneously find that in static tasks other methods outperform the authors' approach.

This reviewer pushes back on the term "data-free RL", as there still needs data (and interaction data) to learn their method.  This is a very confusing terminology, and honestly the generalization comes from the large-scale pretrained VLM - it would be more appropriate to reuse the terminology of zero-shot reward models or zero-shot policy learning used in prior works across alignment methods (vision-language models are zero-shot reward models for reinforcement learning, [Rocamonde, '23]) and diffusion (text-aware diffusion for policy learning, [Luo, '24]).

This reviewer really enjoys the work but believes there are many open questions that warrant further explanation.  Furthermore, the evaluation suite (environments) and comparison suite (benchmarks) is rather weak.  The idea is indeed neat, but the execution leaves much to be desired, and therefore this reviewer believes the work is of borderline quality.

**Questions:**

Why Stickman instead of Humanoid?  Humanoid has been able to be solved with pixel-based world models in the past (Appendix A of Dreamerv2).  What are the specifications of Stickman and with what criteria was it designed?

Why did the authors use a simple GRU?  Why was a more advanced world model not used, like RSSMs?  Was this tested or ablated over?

Why were there no multi-task generalization experiments performed for Meta-World kitchen?

Would the authors consider Text2Reward and other approaches that learn a reward function as data-free RL, as there do not need additional data to generalize to learning new policies?  Alternatively, the data-free RL paradigm sounds like zero-shot generalization for policy learning, which is already offered by VLM-RM and other similar works.

Why did the authors choose to generate the video demonstrations synthetically, rather than use actual natural video clips?  Would the performance not be better when using natural videos which are more in line with what the base VLM was trained on?

How are the rewards computed using temporal alignment?  Essentially, are only the rewards for the most-aligned segments across the target trajectory and the agent used as rewards, and for all other timesteps a 0 reward is provided?  This computation seems rather expensive for long-horizon trajectories.

**Limitations:**

The limitations section seems acceptable.

---

> ### Author Rebuttal · Authors · 2024-08-06
>
> We thank the reviewer for the helpful comments.
>
> **Experiments**
>
> We added new experiments, including stronger model-based baselines. Details are in the main rebuttal message.
>
> We also tested all baselines with LIV's representation in the Kitchen tasks. We used LIV's open-source code to download and instantiate the model. Note: the available model is the general pre-trained model, not the one fine-tuned for the Kitchen environment.
>
> |                   | IQL-LIV     | TD3+BC-LIV  | TD3-LIV     | WM-CLIP-LIV | GenRL       |
> |-------------------|-------------|-------------|-------------|-------------|-------------|
> | kitchen microwave | 0.03 | 0.0    | 0.2   | 0.0    | 0.97  |
> | kitchen light     | 0.53 | 0.05  | 0.0    | 0.0    | 0.46  |
> | kitchen burner    | 0.2  | 0.0 | 0.0    | 0.0    | 0.62  |
> | kitchen slide     | 0.03  | 0.03  | 0.0    | 0.67  | 1.0    |
>
> LIV's results confirm the original paper's claims (see Appendix G4) that the representation does not work well for vision-language rewards without fine-tuning on domain-specific vision-language pairs (which are unavailable in our settings, as we use **no language annotations**).
>
> Text2Reward requires a different experimental setup than ours. We focus on visual RL. The agent observes the environment only through images and doesn't have any privileged information about which entities are present in the scene e.g. joints, limbs, objects.
>
> Normalizing results using expert performance and random performance as max-min values is one of the de-facto standards for the Atari suite. This is referred to as Human Normalized Score (HNS), see [1].
>
> **Aligner**
>
> The aligner is not a generative model and, as the reviewer suggests, it's closer to a denoiser network. In the main rebuttal comment, we have provided an additional explanation about how and why we reduce the multimodality gap by adding noise to the embeddings. Additional insights are available in related work [2,3].
>
> As described in the Appendix (Line 491-492), "the aligner network employs a small U-Net, with a bottleneck that is half the size the embedding representation.".  The embedding representation is 768-dimensional. Further details can be found in our accompanying code implementation, provided to the Area Chair.
>
> **Inverse connector idea**
>
> We have implemented this idea and added it as a baseline (WM-CLIP) to our work. The reviewer's intuition is correct as the idea works. However, without the connecting-aligning process, the system does not perform as well for some prompts/tasks. Please, see the main rebuttal comment for further details.
>
> **Website visualizations**
>
> We added the corresponding task labels in the website and we provided the language prompts in the Appendix.
>
> In the videos, we observe that less natural behavior is correlated with a lower normalized score on the task. Thus, we expect those behaviors to be less smooth, e.g. see 'stickman flipping' or 'walker high kick'.
>
> **Dynamic tasks**
>
> We provided additional results and visualizations, including more dynamic tasks. Please, see the main rebuttal comment and the updated website.
>
> Nonetheless, we agree with the reviewer that the model struggles more to sync with input videos for dynamic actions. One of the reasons for this is that the control frequency of the agent is different from the motion frequency from the provided video prompt. This makes it hard to match the prompt accurately.
>
> **Data-free**
>
> We provided an additional explanation in the main rebuttal comment. We believe that both Rocamonde et al, Luo et al, and Text2Reward fall in the category that we represent as "Offline RL + VLM", as they require data for training the behavior policy.
>
> Given that the term "data-free RL" may be confusing, we have changed our naming convention to "Data-free Policy Learning", similar to Luo et al, which we now cite for reference.
>
> **Humanoid**
>
> We designed the Stickman environment to explore tasks that require upper body limbs (e.g. boxing, doing a handstand) without the complexity of training a humanoid, which requires a significantly larger amount of data to be solved, as reported in the DreamerV2 paper.
>
> Building on the Walker model, the number of joints is increased by 4: 2 joints per arm, one is for the shoulder, the other for the elbow. The total number joints is 10. The action space is normalized to be in [-1,1] as all dm_control tasks.
>
> **Connector architecture**
>
> The architecture of the connector network is the same as the RSSM architecture used for the world model, which employs a GRU for computing the hidden state. We corrected this detail in the paper.
>
> **Additional tasks**
>
> Meta-World environments adopt different environments for different tasks, i.e. with different objects, workspaces and thus visual dynamics. Instead, we focussed on evaluating on domains where we can learn a single MFWM for the environment's dynamics and use it to generalize to new tasks.
>
> The Franka Kitchen environment allows for more tasks in a single environment (we evaluate on 4). However, it's hard to design new tasks as the number of objects to interact with is limited.
>
> **Video prompts**
>
> Natural videos actually work well (and often better) than AI-generated videos. We found intriguing the idea of generating the prompts using another GenAI model, rather than having to look for a video online. In the additional results, we chose to also show some natural videos.
>
> **Temporal alignment**
>
> Before the aligned segments, the reward is computed using the initial state of the target sequence. This is currently stated at Line 155.
>
> We hope to have satisfied all the reviewer's concerns and we look forward to receiving updated feedback.
>
> [1] A Review for Deep Reinforcement Learning in Atari: Benchmarks, Challenges, and Solutions, Fan el al
>
> [2] Mind the Gap: Understanding the Modality Gap in Multi-modal Contrastive Representation Learning, Liang et al
>
> [3] Connect, Collapse, Corrupt: Learning Cross-Modal Tasks with Uni-Modal Data, Zhang et al

---

> ### Comment · Reviewer_Yb6H · 2024-08-13
> **Reviewer Response [1]**
>
> This reviewer appreciates the detailed rebuttal from the authors, and apologizes for the tardiness in response.  Here are provided thoughts in response:
>
> **On experiments:** The additional comparison against LIV is appreciated.  This reviewer understands the point the authors make, that Text2Reward requires a different experimental setup.  However, the project has quite a broad scope: it offers a way to condition on text and perform a behavior, as well as condition on an image (or video) and perform a behavior as well.  In principle, this behavior should be compared against other methods that do this as well, not necessarily isolated to multimodal rewards (which technically the proposed approach does not fall into the category of either).  Therefore Text2Reward would be interesting to show, because it is another technique that converts natural language to a policy (albeit requiring learning - this distinction can be made in analysis).  Furthermore, methods that go purely from image (or video) to policy behavior would be interesting to show - for example, VIP [2] (an image-only ancestor of LIV), or LEXA [3] (which appears remarkably similar to some of the **Video prompts decoded** section of the website) or other goal-conditioned or pose-conditioned RL.  For each capability of the model, it is useful to demonstrate comparisons against existing techniques that perform similar capabilities for completeness; it is indicative of the power of the method the authors are proposing, that there are so many potential capabilities, and this adds greatly to the excitement of the approach - but at the same, in principle, time such capabilities should be thoroughly explored and vetted.
>
> **On evaluation:** This reviewer is aware that HNS is a standard evaluation technique for Atari, but was under the impression that the purpose of such a technique was to measure *superhuman performance* (indeed, this is suggested as much in Subsection **Human Average Score Baseline** of [1], which the authors have linked).  It seems strange to apply it to an expert policy, which is still just a policy and not a human.  Under such circumstances, where every policy is equally a policy, why does HNS-style evaluation still make sense rather than just raw episode rewards?  This reviewer has not seen this evaluation technique outside Atari; it definitely does not seem common practice for reporting DeepMind Control Suite performance (raw rewards), nor for FrankaKitchen (success rate).  From this reviewer’s perspective, it does not seem to detract from the story to report results in the standard way, and it is puzzling that HNS was chosen for these instances without strong justification.
>
> **On the Aligner:** The clarification in the rebuttal PDF as well as the response is very useful, and appreciated.
>
> **On the Inverse Connector:** This reviewer appreciates the additional results, and the preliminary insights are very interesting.  Provided the final draft includes these results paper, as well as analysis into the performance discrepancy based on projection direction, this reviewer is willing to increase their rating.  This reviewer believes examining and explaining the projection direction would strengthen the connection between VLM-RM dynamics and in-domain world modeling dynamics, and the paper would benefit from featuring it in the main text.
>
> **On updated website visuals:** This reviewer is confused by the note that “less natural behavior is correlated with a lower normalized score on the task. Thus, we expect those behaviors to be less smooth, e.g. see 'stickman flipping' or 'walker high kick'.”  There does not seem to be a score for such tasks that could be referred to as low; nor should there be, since they are novel behaviors.  To what are the authors referring to?  Also, apologies for the poor memory - but this reviewer really cannot recall the original website examples, and therefore cannot meaningfully determine the delta; in future updates would the authors please prominently demarcate the delta (perhaps with different colors/fonts/highlightings)?
>
> **On Dynamic Tasks:** This reviewer agrees with the analysis that control frequency and motion frequency mismatch is a difficult consideration to overcome, and can potentially contribute to a preference for stationary policies.
>
> [1] A Review for Deep Reinforcement Learning in Atari: Benchmarks, Challenges, and Solutions, Fan, 2021.
>
> [2] VIP: Towards Universal Visual Reward and Representation via Value-Implicit Pre-Training, Ma, 2022.
>
> [3] Discovering and Achieving Goals via World Models, Mendonca, 2021.

---

> > ### Comment · Reviewer_Yb6H · 2024-08-13
> > **Reviewer Response [2]**
> >
> > **On Data-free:** The name “Data-free Policy Learning” does seem more appropriate, as subsequent policies no longer need data to be trained (the data is only used to learn a world model), and can be done in imagination alone.  It is totally fine under scrutiny, but also sounds strange on first hearing - “data-free” and “learning” seem like an oxymoron.  Perhaps “Policy Learning in Imagination”?  Or “Data-Free (Policy) Generalization through Imagination”?  The term could definitely be workshopped, but this is not a critical matter.
> >
> > **On Humanoid:** This reviewer agrees that Humanoid is more complex; however, learning the world model is a *one-time fixed cost*, on top of which arbitrary new policies can be learned.  Would learning new policies in imagination still incur a high cost?  Intuitively, the world model should decrease training cost because planning is possible.
> >
> > **On the Connector:** Sure.
> >
> > **On Multitask Generalization:** Apologies, the reviewer meant to refer only to FrankaKitchen.  Regardless, it would be interesting to train the world model on a subset of available tasks and have a holdout novel task.  Multitask generalization capabilities for a robotics environment would be very exciting to demonstrate, even if for a simpler setup.  Alternatively, simply showing a novel task generalization for the robotic arm (not necessarily tied to any task), like what was demonstrated for Stickman in the website, would also be very interesting to show.
> >
> > **On Natural Video Prompts:** Sure.
> >
> > **On Temporal Alignment:** This certainly seems hacky, but not a critical issue of the work.
> >
> > Ultimately, this reviewer appreciates the updated experiments, explanations, and insights (e.g. the projection direction), and is willing to tentatively upgrade the score in favor of acceptance.  However, there still remain reservations about the thoroughness and completeness in verifying all capabilities of the proposed approach, as well as a thirst for more results in the domains chosen (e.g. FrankaKitchen/robotics multitask generalization, and as a [non-critical] reach, some Humanoid demonstration).  Further clarification and justification would also help (e.g. the choice of HNS when humans are not involved, which to this reviewer's knowledge is a rare design decision).

---

> > > ### Author Response · Authors · 2024-08-13
> > >
> > > We would like to thank the reviewer for the extensive feedback provided. We are glad the additional material was useful and improved the reviewer's opinion of our work.
> > >
> > > ## On experiments
> > >
> > > > Text2Reward requires a different experimental setup. However, the project has quite a broad scope: it offers a way to condition on text and perform a behavior, as well as condition on an image (or video) and perform a behavior as well.
> > >
> > > We could not find any experiments or examples in the Text2Reward paper where they condition on images. The closest we found on the paper is, in their Figure 6, the experiments were the user provides a feedback on the observed rollouts. However, this requires a human in the loop looking at the rollout and providing verbal feedback to the system. This is currently out of scope for our work and might be considered in future extensions.
> > >
> > > We definitely agree with the reviewer that additional baselines are useful in assessing the generality of an approach. However, in order to provide a fair comparison, it is important that we benchmark all the baselines in the same settings. If certain baselines require significant changes to the data used, the inputs to the system, or the foundation models adopted, it becomes harder and harder to compare methods in a fair way.
> > >
> > > For the **Behaviors from video prompts** experiments, all the baselines adopt the same training data, the same foundation model to process the videos (InternVideo2), the same video prompts. The only difference between them is the underlying algorithm to specify rewards and the policy learning method adopted. These are our main contributions and thus these are the aspects we would like to compare in these experiments.
> > >
> > > ## On evaluation
> > >
> > > > This reviewer is aware that HNS is a standard evaluation technique for Atari, but was under the impression that the purpose of such a technique was to measure superhuman performance (indeed, this is suggested as much in Subsection Human Average Score Baseline of [1], which the authors have linked
> > >
> > > The reference provided states two motivations for the HNS: one is mentioned by the reviewer, the other is "Performance across algorithms become comparable. Like Max-Min Scaling, the human normalized score can also make two different algorithms comparable."
> > >
> > > We have provided Atari as an example for (human) normalized score, as this is one of the most common benchmarks. However, max-min scaling between expert and random performance, or more simply, max scaling by expert performance, is very common in the literature. We provide some additional references as follows:
> > >
> > > - *A Generalist Agent, Reed et al, 2022.* From Figure 5's description: "Here values on the x-axis represent a specific percentage of expert score, where 0 corresponds to random agent performance"
> > >
> > > - *URLB: Unsupervised Reinforcement Learning Benchmark, Laskin et al, 2022.* From Figure 3: "Scores are normalized by the asymptotic performance on each task (i.e., DrQ-v2 and DDPG performance after training from 2M steps on pixels and states correspondingly)."
> > >
> > > - *TD-MPC2: Scalable, Robust World Models for Continuous Control. Hansen et al, 2024*. All the results are provided using "Normalized scores" (the way this is computed is not clearly stated in the paper but we assume it's either expert performance or the maximum achievable performance on each task)
> > >
> > > - (ProcGen) *Leveraging Procedural Generation to Benchmark Reinforcement Learning. Cobbe et al, 2020*. From page 3: "For each environment, we define the normalized return to be Rnorm = (R − Rmin)/(Rmax − Rmin), where R is the raw expected return and Rmin and Rmax are constants chosen to approximately bound R."
> > >
> > > The reason why it's so important to normalize returns/scores is because different tasks have different performance scales. For instance, the maximum return in Cheetah Run is around 850, for Quadruped Run is around 650, for Walker Walk is around 970. For the Kitchen tasks, since we follow the original paper [1] in using success rate as a metrics, the possible scores are only 0 and 1.
> > >
> > > These large differences in the performance scales require us to use a way to normalize scores and Max-Min scaling is a very common way in the literature to do so, when multiple different tasks are involved.
> > >
> > > [1] Relay Policy Learning: Solving Long-Horizon Tasks via Imitation and Reinforcement Learning, Gupta et al, 2019
> > >
> > > ## On the Inverse Connector
> > >
> > > We thank again the reviewer for providing the idea for these additional experiments and we are glad the additional insights are found to be useful.

---

> > > > ### Author Response · Authors · 2024-08-13
> > > >
> > > > ## On updated website visuals
> > > >
> > > > We will try to clarify our statements referring to the website visuals. We remind that a normalized score of 0 = random performance, while 1 = expert performance (i.e. Dreamer trained to solve the task using a hand-designed reward function and achieving smooth behavior).
> > > >
> > > > **Walker Walk**: GenRL's performance on this task is 1.01 normalized score. GenRL is performing as well as an expert agent. The behavior shown on the website is indeed quite smooth.
> > > >
> > > > **Stickman Run**: GenRL's performance on this task is 0.35 normalized score. We observe that the agent is indeed running as requested, but the behavior is a bit awkward: the agent tends to fall and the running movement is not very smooth.
> > > >
> > > > **Stickman Lunge Pose**: GenRL's performance on this task is 0.80 normalized score. The behavior looks correct and mostly smooth, but the agent still oscillates around the lunge pose, not achieving perfect score.
> > > >
> > > > **Walker High Kick**: GenRL's performance on this task is 0.48 normalized score. We observe that the agent is indeed kicking as requested, but the behavior is not perfect: the agent tends to fall on the back or on its knees.
> > > >
> > > > There is a correlation between the agent's score on a certain task and how smooth the behavior produced looks. Full results per each task are available in the Appendix of the paper.
> > > >
> > > > ## On Data-free
> > > >
> > > > We think "Data-Free (Policy/Behavior) Generalization through Imagination", as proposed by the reviewer, it's a good fit for the paragraph.
> > > >
> > > > ## On Humanoid
> > > >
> > > > We agree that training the world-model is a one-time cost. However, there are many other computation costs to consider.
> > > >
> > > > In order to add experiments on the Humanoid environment, we need to collect both expert data and exploration data. We used Dreamer to collect expert data in our experiments. If we would like to keep the same setup, since Dreamer takes around 30M frames to converge in the Humanoid Walk, collecting the data would likely require more than 300 hours (see [2]). This is only for the Walk task, the Run task would probably require longer, and the exploration data collection might also take a very long time, if we would like a similar amount of exploration data.
> > > >
> > > > After data collection, we would still need to pre-train the world model . This will potentially take longer than 5 days, and will potentially require a larger model, given the significantly larger amount of data (one order of magnitude larger). A similar issue will apply to the baselines, which may also take much longer than 7 hours to train.
> > > >
> > > > We agree that having the Humanoid environment would be a great addition. However, the costs for having this additional environment are definitely high and represent a barrier.
> > > >
> > > > [2] Mastering Visual Continuous Control: Improved Data-Augmented Reinforcement Learning, Yarats et al
> > > >
> > > > ## On Multitask Generalization
> > > >
> > > > Adding additional tasks to the Kitchen environment, in offline RL (or data-free) setting is harder than for the other environments. This is due to the difficulties of exploring meaningful behaviors in manipulation environments.
> > > >
> > > > We are able to use GenRL to retrieve the four tasks present in the dataset and we are generally able to achieve similar tasks, such as "reach the microwave". From decoding the language prompts, we also see that GenRL can sensibly decode prompts such as "moving to the left" or "staying still", or more 'unusual' prompts such as "swan pose" (where the robot arm would imitate the pose of a swan). While these prompts show that the system works well in this environment, we found that none of these tasks would be particularly useful or worth showing.
> > > >
> > > > When we prompt the system with more interesting tasks that are out of the distribution, the agent generally struggles. For instance, if we ask to open the double door cabinet on the top left, the agent may reach for it but will not open it. The reason behind this is that it probably never opened that cabinet in the training dataset and thus it's impossible to reproduce such behaviors in an offline manner. The same issues are most likely present with all the baselines as well, as they all use the same dataset and foundation models for training. For the locomotion domains, instead, the exploration data is varied enough that new tasks, outside of the tasks present in the training dataset, are enabled through the exploration data (see the ablation study on the Walker, where the exploration data is shown to be very useful in improving multi-task generalization),
> > > >
> > > > We could add these details to our discussion if the reviewer finds them useful.
> > > >
> > > > We would like to thank the reviewer once again and we hope this comment clarifies the reasons behind our work and our experimental choices further. We look forward to any additional feedback.

---

### Official Review · Reviewer_bBQY · 2024-07-16

**Soundness:** 2
**Presentation:** 3
**Contribution:** 3
**Rating:** 7
**Confidence:** 3

**Summary:**

The paper looks at a method for leveraging foundation multimodal models for learning world models in RL. They do so by aligning the latent space of a video language model with that of a generative model that can be used for learning in imagination. This is done by training connector-and-aligner networks . The rewards for a task can then be derived by measuring the cosine sim between representations of the states visited by a policy and the states generated by the connector aligner network when it is conditioned on a language-based task prompt. A policy can be optimised to maximise this alignment based reward.

**Strengths:**

Transferring foundation model knowledge to improve policy learning is an open problem of interest to the community.

The paper provides a successful recipe for aligning a foundation model with the world model for a specific domain that we want to do policy learning in.

The paper is written well.

I'm currently being conservative in giving a borderline accept score, since some aspects of the method are not clear to me (I have addressed this in my questions below) - but I will be happy to raise my score after engaging with the authors once they have addressed these questions.

**Weaknesses:**

1. I would have expected that simple tasks with clearly distinguishable static end states (such as standing) should have worked equally well with CLIP rewards, however the table shows a big difference between the proposed method and the image-language reward baselines even on those tasks, which leads me to think that the baselines may be missing out on some component that the proposed method has. What could be missing, or is this intuition wrong?
2. The generations in Fig 6a are actually not accurate at all - many of the poses don’t correspond to the humanoid pose if you look closely and would actually optimize learning to strike the wrong pose if a policy is trained with it.

**Questions:**

1. Why is setting b=imagination horizon the same as doing no alignment? (Line 160)
2. I’m not completely sure how you train the aligner-connector network: is it done by 1) using images collected from the downstream domain (in this case from the mujoco sim), 2) getting their VLM visual embedding and their world model encoder embedding and aligning those? As for the text part, is this done by corrupting the VLM visual embedding (to approximate the language embedding) and aligning it again with the world model encoder?  What is the policy used to collect the data and resulting data distribution? I understand that Fig 5 is somehow related to this question but this could be made clearer. For eg. which task’s policy is chosen to collect the data to train the MWFM for the results in the main table (how is this policy related to the task being evaluated)?
3. The discussion around Figure 5 is not very clear to me - how do we infer that “’run’ data proves more generalizable than  ’walk’ or ’stand’ data across tasks“ - the figures suggest that training on ‘stand’ led to the highest rewards for downstream tasks
4. “This can be explained by  the fact that the target sequences that GenRL infers from the prompt are often slightly in motion“ - could you explain why that would be the case (it inferring the closest matching state as one that is in motion)?

**Limitations:**

The paper includes a brief discussion on limitations of their method.

---

> ### Author Rebuttal · Authors · 2024-08-06
>
> We thank the reviewer for the helpful comments.
>
> **Image-language CLIP results**
>
> > simple tasks with clearly distinguishable static end states (such as standing) should have worked equally well with CLIP rewards
>
> We agree with the reviewer's intuition and we believe the results confirm their statement. If we look only at the static tasks (stand tasks and kitchen tasks) we see that the performance of Image-based and Video-based baselines are very comparable. We observe this also for the new model-based baselines (please, see main rebuttal message). GenRL, which uses a video-language VLM, tends to outperform the other approaches.
>
> **Video prompts' poses**
>
> The visualizations in Figure 6a are the interpretations of what the visual prompt provided would look like in the embodied domain, according to the model. We agree these do not match the poses provided exactly, but only in their "semantic" meaning (e.g. doing crunch abs, despite the legs position being different). This issue is due to two aspects of our framework: (i) VLMs shine with semantic interpretation but struggle to represent precise geometrical information, and thus they struggle with precise pose estimation, (ii) the world model may have not seen certain poses in its training dataset, and thus it will provide the closest behavior to the prompt. Despite this limitation, we found that our video prompting allows to learn behavior from a variety of visual conditions, e.g. draw sketches, very different camera views. Please, see our main rebuttal message for the additional experiments.
>
> **Sequence alignment**
>
> If we set $b$=imagination horizon we only have one possible comparison between the two sequences.
>
> Example with a sequence of length 3. Target sequence states (T), agent sequence states (A).
>
> b = 1 (initial state alignment), 3 possible alignments
>
> | T | |   |
> |---|---|---|
> | A | A | A |
>
> |  | T |   |
> |---|---|---|
> | A | A | A |
>
> |  | |  T |
> |---|---|---|
> | A | A | A |
>
> b = 2, 2 possible alignments
>
>
> | T |T |   |
> |---|---|---|
> | A | A | A |
>
> |  | T | T  |
> |---|---|---|
> | A | A | A |
>
>
> b = 3 (no alignment), 1 possible alignment
>
> | T |T |  T |
> |---|---|---|
> | A | A | A |
>
> **Training procedure and data collection**
>
> We think the reviewer correctly understood our training procedure, and which we further clarify in our main rebuttal comment for clarity. As currently stated in Lines 174-176, we collect data using two kind of agents: (i) a Dreamer V3 agent learning to perform the task (we added the details about the agent being DreamerV3 in the main text) and (ii) a Plan2explore agent, collecting exploration data in the domain.
>
> The datasets are detailed in Appendix. However, we provided detailed results per task on the behavior extraction tasks so that a reader can quickly see which tasks are contained in the dataset (indeed, the behavior extraction tasks). We have made this clearer in the main text, by stating it at Line 190 as follows:
>
> "Behavior extraction. We want to verify whether the methods can retrieve the tasks behaviors that are
> certainly present in the dataset. *For these tasks, the replay buffer of an expert agent learning to solve the tasks (DreamerV3) was already included in the mixed dataset used for training*"
>
> **Training data distribution**
>
> We believe there is a misunderstanding in the way Figure 5 visualizes the data. The labels (all, expl, run walk, stand) indicate the portion of the data used for training the agent. Each plot represents performance of the agent trained with different portions of data on a task / group of tasks. Thus, the agent trained only on 'stand data' is the one with the lowest performance. We changed the labels to "all/expl/run/walk/stand **data**" to improve clarity.
>
> **Static poses matching**
>
> Examples of predicted sequences being in motion for static tasks are present in our website. For example, if we look at the decoding of the "downward facing dog" prompt, we observe that the pose of the agent is in motion, rather than being completely static. We observed similar issues with some of the task prompts.
>
> We hope to have satisfied all the reviewer's concerns and we look forward to receiving updated feedback.

---

> > ### Comment · Reviewer_bBQY · 2024-08-10
> >
> > Dear Authors,
> >
> > Thanks for your response, I have now read through the responses and the pdf.
> >
> > My impression is that the connector-aligner is doing a lot of the heavy lifting (not just in terms of more trainable parameters but by adding more flexibility to the language representations since they are processed by a denoiser). Is there a way to compare to the CLIP rewards more fairly then, for eg. by training an aligner in a similar fashion on top of the CLIP visual embeddings and then using the image/text encoders + the aligner at test time? That will help to really decouple where the performance gap is coming from.

---

> > > ### Author Response · Authors · 2024-08-12
> > >
> > > Dear Reviewer,
> > >
> > > We want to thank you for your feedback.
> > >
> > > We agree that the connector-aligner mechanism, which along with GenRL's reward function represents our main contribution, is crucial in our work.
> > >
> > > As requested by the reviewer, we ran additional experiments on all 35 tasks (behavior retrieval + multi-task generalization benchmarks) to establish the importance of the **aligner** network.
> > > In the following table, we report the results of two additional methods:
> > > - **GenRL - no aligner**: this is an ablation of GenRL where the the language prompt's embedding is directly fed into the connector, rather than processing it first with the aligner.
> > > - **TD3-V and WM-CLIP-V + aligner**: for these baselines, we first process the language prompt's embedding using GenRL's pre-trained aligner. Then, we use it to compute the cosine similarity for the reward function, as for the original baselines.
> > >
> > > We summarize the results by domain in the following table (2-3 seeds to increase for the camera-ready):
> > > |           | GenRL - no aligner | GenRL        | WM-CLIP-V    | WM-CLIP-V + aligner | TD3-V        | TD3-V + aligner |
> > > |:---------:|:------------------:|:------------:|:------------:|:-------------------:|:------------:|:---------------:|
> > > | quadruped (6 tasks) | 0.17  ± 0.02       | 0.90  ± 0.02 | 0.81  ± 0.04 | 0.76  ± 0.05        | 0.32  ± 0.04 | 0.33  ± 0.04    |
> > > | walker (9 tasks)    | 0.19  ± 0.01       | 0.75  ± 0.01 | 0.70  ± 0.02 | 0.74  ± 0.01        | 0.56  ± 0.04 | 0.48  ± 0.04    |
> > > | stickman (13 tasks)  | 0.09  ± 0.01       | 0.66  ± 0.01 | 0.54  ± 0.03 | 0.50  ± 0.03        | 0.38  ± 0.02 | 0.38  ± 0.02    |
> > > | cheetah (3 tasks)   | 0.32  ± 0.02       | 0.93  ± 0.01 | 0.82  ± 0.17 | 0.84  ± 0.02        | 0.31  ± 0.09 | 0.77  ± 0.04    |
> > > | kitchen (4 tasks)   | 0.25  ± 0.00       | 0.76  ± 0.08 | 0.71  ± 0.14 | 0.84  ± 0.09        | 0.32  ± 0.16 | 0.27  ± 0.10    |
> > > | overall (35 tasks)   | 0.17 ± 0.00        | 0.76 ± 0.01  | 0.67 ± 0.03  | 0.68 ± 0.02         | 0.40 ± 0.02  | 0.42 ± 0.02     |
> > >
> > > We can observe that:
> > > i) the aligner mechanism is crucial in GenRL's functioning.
> > > ii) processing the language embedding in the reward function of the WM-CLIP-V and TD3-V baselines changes performance on some tasks (performance per domain varies). However, using the aligner provides no advantage overall.
> > >
> > > **Intuition behind the results**
> > >
> > > We believe the aligner is very important in GenRL because its output, the processed language embedding, is fed to another network, the connector. If the language embeddings were not processed by the aligner, they would have been too different from the embeddings used to train the connector, which are the visual embeddings. We provide an additional explanation for this in our main rebuttal comment.
> > >
> > > Instead, for the baselines, we process the language embedding with the aligner and then use it to compute a similarity score with the visual embeddings. This overall renders very similar performance to no aligner processing, hinting that the aligner network doesn't improve the cosine similarity signal. At the same time, this also suggests that the aligner network doesn't hurt the generality of the VLM's embeddings, as the cosine similarity after processing the embedding provides a similarly useful signal as before processing.
> > >
> > > We hope this provides additional insights into our work and we look forward to receiving additional feedback.

---

> ### Comment · Reviewer_bBQY · 2024-08-13
>
> Thank you for your response, and I agree that the connector-aligner is a contribution of the work.
>
> The additional experiments in the rebuttal help to disentangle the contributions of the different components but it is still not clear to me why GenRL without aligner is so much worse than the baselines when they don't even benefit from the aligner?
>
> My guess for where the advantage of GenRL is coming from, would be that the connector network is basically allowing you to go beyond the performance of VLM-RM style methods by adding flexibility to how the similarity is taken (for eg. maybe some dimensions of the VLM representation are not informative or important when computing the similarity for rewards, and they are adding noise to the similarity scores, then the connector network allows you to ignore these dimensions when transforming them to a different representation space?).
>
> Regardless of the exact reason, (which I hope the authors will do more digging into and analyze for the camera ready), I think the proposed additions are allowing the proposed method to extract/transfer concepts from VLMs to a greater extent than prior work, for learning control policies. Thus I will update my score to reflect this.

---

> > ### Author Response · Authors · 2024-08-14
> >
> > > The additional experiments in the rebuttal help to disentangle the contributions of the different components but it is still not clear to me why GenRL without aligner is so much worse than the baselines when they don't even benefit from the aligner?
> >
> > As the reviewer suggested later in their comment, GenRL's performance advantage might be attributed to its policy learning being driven by a latent features similarity, rather than by the similarity of the CLIP embeddings.
> > Without the aligner, GenRL fails (almost completely) to translate language embeddings into world model states. Thus, the similarity of the latent features is not meaningful for solving the task at hand.
> >
> > We can provide additional visualizations about this in the Appendix for the camera-ready, where we show the difference between decoded language prompts with and without using the aligner. Without the aligner, the decoded prompt sequences tend to not follow the given language prompt. From this addition and the ablation study performed, it should be clear that the aligner's use is bridging the gap between the two modalities (language and vision) for the connector, rather than providing "denoised" CLIP embeddings.
> >
> > We would like to thank the reviewer for their valuable feedback, which has significantly improved the quality of our work. The additional ablation study and the clarifications we incorporated based on the reviewer's recommendations have improved our presentation.
> >
> > We are also very pleased to see that our rebuttal positively influenced the reviewer's opinion of our work, leaning towards a firm acceptance recommendation.

---

### Official Review · Reviewer_jfoT · 2024-07-18

**Soundness:** 1
**Presentation:** 3
**Contribution:** 3
**Rating:** 3
**Confidence:** 4

**Summary:**

This paper proposes to combine a DreamerV3-style world model with a pretrained vision language model (VLM). By training two small adaptors to align the latent space of the VLM with that of the world model, the aligned representations from the VLM can be used as a reward signal to train agents in the world model.

The training process consists of two main parts.

1) There is a large offline dataset needed in the environments of interest (prompt and trajectories of states and actions), generated by expert RL agents and a random policy. This trains the world model and the adaptors. Each environment (domain) uses a separate world model.

2) Actor-critic agents are trained purely within this world model’s imagination, a separate policy for each task. The paper shows these agents outperform standard model-free offline RL methods trained on only the large offline dataset. It also shows some effectiveness at generalizing to new tasks within an environment, specified with a new text prompt.

**Strengths:**

- The core idea of the paper is very nice. There is a lot of interest from the community in working out how to get value from the broad general knowledge locked away in LLMs and VLMs, into RL agents. This paper offers a novel way to attack this – to my knowledge world models have not been used in this context before.

- The results are not dazzling, but they indicate the approach works and it outperforms standard (though perhaps weak) offline-RL baselines. Section 4.2 shows promise in generalizing to knew text prompts in an existing environment.

**Weaknesses:**

My main criticism of the paper is that the narrative oversells what the core work actually supports. I detail examples below. Overall I’d suggest either presenting the work that has been done comprehensively in a more reserved manner, or adding the required work to support the broader claims and experiments. Either way, I think changes would be large enough to require a resubmission.  I’m disappointed to not be able to give the paper a higher score as I liked the main idea.
- The capability of the model to condition on visual goals is presented as a main functionality of the model – featuring in the first figure, the abstract, and throughout the paper. But the only evidence to support this is a very brief and qualitative experiment (Figure 6a). Everything else is conditioned on text. I am of the opinion that conditioning on visuals would likely work, but the paper must present good evidence to support this.
- Several aspects of the title ‘Multimodal foundation world models for generalist embodied agents’ are misleading. 1) Only one modality is really tested (as in prior point). 2) ‘Foundation world models’ suggested I'd see a single very general world model. But in Appendix D is an important detail -- each environment learns a separate world model, so they are only general or foundational within a specific Mujoco embodiment. This kind of detail is important and should be honestly discussed in the main paper. 3) A ‘generalist agent’ is referred to, but every agent in the paper only performs a single specialist task, there is nothing general about the agent’s themselves.
- The method is reported as needing ‘no language annotations’ (line 42). This is not true. The large offline dataset requires text prompts accompanying each trajectory.
- The paper claims to be ‘the first large-scale study of multitask generalization from language in RL’ (line 165), but I can think of others. Language table is the first that comes to mind.
- One of the motivations for the work is that reward functions can be hard to specify, while language is a more natural form. However, the large offline dataset is generated by using multiple expert agents which need reward functions.
- ‘Data free RL’ is suggested as a new paradigm for foundation models in RL. I’d argue that this is simply know as zero-shot generalization to most in the community.
- Main experiments are presented in Table 1. Whilst the offline-RL methods are one comparison point, I’m not sure how comparable they are, since they are all model-free while GenRL is model based. Are there any model-based variants that would be easily considered as baselines? The differences are reflected in the different compute times required – GenRL takes 5 days for world model training +5 hours per policy, while the baselines take 7 hours per policy. This seems like an unfair comparison, especially to withhold the detail to the appendix.
- Results in Minecraft are briefly mentioned in Section 5. But so few details are given that I am lost as to what it is showing. This should either be removed or full details added.
- The paper presents a new stickman environment. But details are sparse. The authors have failed to correctly identify this in Checklist Section 13.

**Questions:**

See weaknesses.

**Limitations:**

Fine.

---

> ### Author Rebuttal · Authors · 2024-08-06
>
> We thank the reviewer for the helpful comments.
>
> **Learning from visual prompts**
>
> In order to support our claims with empirical evaluation, we have provided results of behavior learning from video prompts. The results can be found in our main rebuttal message and the videos on the website.
>
> We have three main observations about these experiments: (i) GenRL performs overall better than the baselines, (ii) the performance from video prompts are generally close to the performance from language prompts, (iii) leveraging the foundation VLM knowledge allows to generalize in very different visual settings.
>
> **Title**
>
> We hope to have resolved the reviewers' concern about the multimodality nature of our work with the additional experiments.
>
> The term we coined as a shortand for our MFWM models is not used to indicate a class of foundation (world) models. Instead, as we state several times in the paper (Lines 9, 42, 105, Figures 1 and 2), it is used to indicate world models whose representation is connected to the knowledge of multimodal foundation models. In order to make this more clear, we added an hyphen to our name ("Multimodal-foundation world models") so that it is clear that the first two words are used as an "adjective" for the world model, rather than for indicating a multimodal foundation model.
>
> We understand the definition of a "generalist agent" may be ambiguous, as we have a "generalist" model that allows multi-task and multimodal prompts generalization by training multiple specialist policies, as correctly stated by the reviewer.
>
> Given the above observations, we will update our title to:
> **"Multimodal-foundation world models for generalization in embodied domains"**
>
> Moreover, following the reviewer suggestion, we moved the statement about training one world model per dynamic, currently in the Appendix, to the main paper (Experiments section).
>
> **No language annotations**
>
> There must have been a misunderstanding, probably due to an ambiguous statement at Lines 176-177. However, we use **no language annotations** in our datasets. Please, see our main rebuttal comment about this.
>
> **Large-scale study**
>
> To the best of our knowledge, our work is the first large-scale study where agents can generalize to as many embodied tasks and domains (35 tasks, in 5 embodied domains). Nonetheless, we will remove the adjective "first", if the reviewer finds it inaccurate.
>
> **Data collection**
>
> As we developed our framework, we observed that, in order to solve more complex tasks, the agent requires some expert data/demonstration of the complex behavior. We analyse this in our "Training data distribution" experiments. We believe this limitation is, to some extent, inevitable, as data-driven deep learning agents need to observe complex behaviors during training in order to be able to replicate them.
>
> In this work, we used an expert RL agent (DreamerV3) to collect the data for us. Using a small set of demonstrations might be an alternative. We added further discussion about this and other limitations in the Appendix of the paper.
>
> **Data-free**
>
> We discuss the differences between Offline RL, GenRL with data and GenRL data-free in our main rebuttal comment.
>
> Generally, we would argue that a single offline RL policy cannot generalize to new tasks zero-shot, without (re)training on the data. Skill-based approaches hold the potential to do so. However, training skill-based policies that cover many different complex behaviors in a zero-shot fashion is an open research question. Our data-free strategy offers an alternative view on this problem.
>
> **Comparison with baselines**
>
> As we describe in the main rebuttal message, we added model-based baselines to our main experiments. These baselines follow the recommendation of Reviewer Yb6H of learning an "inverse connector" from the world model representation to the VLM representation (GenRL does the opposite). Please see the main rebuttal message for further details.
>
> GenRL pretrains a single world model per dynamic, for 5 days, and uses it for learning tasks in imagination, for 5 hours per task. Model-free RL requires only 7 hours, as it takes longer to converge, but has no pre-training stage.  On a single GPU, model-free RL is faster to train for a small number of runs. GenRL becomes advantageous when using the world model for training policies for more than 60 runs (runs = tasks x seeds). If using data-free learning (3 hours per task), the advantages become significant after 30 runs.
>
> Given the nature of our work, focusing on behavior generalization performance, rather than on computing budget, and the limited space for the main text, we have kept this information (now including the above discussion) in Appendix.
>
> **Experimental settings**
>
> To make room for the new visual prompted experiments, we are moving the Minecraft results in Appendix. We have also added more details about the training setup we adopted. As we stated in Line 295, we used DreamerV3 to collect the data and thus used the same settings as their work. Nonetheless, we also summarized the main information in our work.
>
> The Stickman environment is based on the Walker environment from the _dm_control_ suite. We designed the Stickman environment to explore tasks that require upper body limbs (e.g. boxing, doing a handstand) without the complexity of training a humanoid (which requires a significantly larger amount of data to be solved [1]). The number of joints is increased by 4: 2 joints per arm, one is for the shoulder, the other for the elbow. The total number joints is 10. The action space is normalized to be in [-1,1] as all _dm_control_ tasks. The robot also presents an head, to resemble a humanoid.
>
> Further details can be found in our accompanying code implementation, which we shared with the Area Chair.
>
> [1] Mastering Atari with Discrete World Models, Hafner et al, 2020
>
> We hope to have satisfied all the reviewer's concerns and we look forward to receiving updated feedback.

---

> > ### Comment · Reviewer_jfoT · 2024-08-12
> >
> > Thanks for the response -- I appreciate the large amount of work and effort that went into the rebuttal. At this point my feeling is that properly interrogating all the new material would best be done in a fresh round of reviews when the changes are integrated into the main paper, though naturally I will discuss this with other reviewers.
> >
> > I did want to clarify the point on the language annotations. I'd assumed the prompts listed in Table 3 were used at training and test time -- i.e. _trajectory-level_ annotations were not used but _task-level_ annotations were. Is this not the case?

---

> > > ### Author Response · Authors · 2024-08-12
> > >
> > > We would like to thank the reviewer for acknowledging our rebuttal.
> > >
> > > > I did want to clarify the point on the language annotations. I'd assumed the prompts listed in Table 3 were used at training and test time -- i.e. trajectory-level annotations were not used but task-level annotations were. Is this not the case?
> > >
> > > We would like to confirm this is not the case. As we detailed in our rebuttal, no text annotations or text embeddings have been used during training of the MFWM components, neither at the trajectory level nor at a task level. We only used visual embeddings for the training of the connector and aligner.
> > >
> > > The prompts listed in Table 3 are not annotations, as they are not associated with any data. These are the language prompts that are used to specify the tasks to solve (is this what the reviewer indicates as "test time"?). As done in previous work, for all methods, these prompts are only used to compute the rewards for policy learning. In the baselines, the reward is computed by obtaining the cosine similarity between the prompt's embedding and visual embeddings, similar to what is described in [1]. In GenRL, the prompt's embedding is used to generate latent targets (using the connector and aligner), to be achieved by leveraging Equation 3.
> > >
> > > [1] Vision-Language Models are Zero-Shot Reward Models for Reinforcement Learning, Rocamonde et al, 2024
> > >
> > > >  At this point my feeling is that properly interrogating all the new material would best be done in a fresh round of reviews when the changes are integrated into the main paper, though naturally I will discuss this with other reviewers.
> > >
> > > During the rebuttal period, we have done our best to provide all the reviewers with the material and the information they requested. Following the conference guidelines, we have provided a single-page PDF to share the results of the experiments the reviewers have asked us to add to our work. We have also provided additional Figures to aid the reviewers' understanding of parts of our work which seemed less clear.
> > >
> > > Overall, the only new material to be added to the main text is represented by the experiments from the visual prompt that this reviewer requested. The other experiments only require adding two columns (the new baselines) to existing sets of experiments. The additional Figures will go in the Appendix for clarification, if the reviewers found them useful (so far we received no feedback on them).
> > >
> > > We want to thank the reviewer once again, as their detailed review has been very useful in improving the clarity of our work and in stating our claims more firmly, thanks to the additional empirical evidence.
> > >
> > > We would strongly appreciate it if the reviewer could provide us with updated feedback on our work and let us know if the material provided and the proposed edits (e.g. in the title) improved the reviewer's opinion about our work.

---

### Official Review · Reviewer_X1kr · 2024-07-18

**Soundness:** 3
**Presentation:** 1
**Contribution:** 3
**Rating:** 5
**Confidence:** 4

**Summary:**

The paper wants to leverage the large-scale pre-training of foundation models trained on internet data to train a world model for embodied agents that generalizes across tasks and domains. This is done by training a world model in the standard way, but in addition training aligner and connector networks that (1) map language embeddings to video embeddings and (2) map video embeddings to world model latent states. At inference time, this allows conditioning the world model on a task language prompt and then training in imagination to learn policies.

**Strengths:**

- On the website, the reconstruction results from language and video are nice and quite unexpected (I'm unsure why the aligner and connector networks are able to generalize to new prompts)
- The problem the paper is trying to solve is relevant, especially given the mismatch in data availability between embodied and vision / language settings

**Weaknesses:**

- The main claim of the paper is strong generalization performance, leveraging the internet scale pre-training of video-language models. The bottleneck is the generalization ability of the networks which map embeddings from the video-language model to the world model latent states, and on the quality of the world model itself. I don't see why the aligner and connector should generalize.
- Given the main claim, I would like stronger baselines / ablations in the generalization and data-free settings. Currently, there are no baselines in the data-free case which makes it impossible to assess how well the method generalizes.
- Many of the experimental details are unclear in the paper (please see my questions). I encourage the authors to explain these better in the rebuttal and camera-ready, and also provide some intuition for why their method is better than the baselines.
- In the single task, offline RL case, all the baselines are model-free, whereas the proposed method utilizes a model. I would have liked to see at least one model-based baseline to confirm that the improvement is because of the better reward signal and not because of the model-based optimization.
- In the single task, offline RL case, reward is computed by looking at the similarity between the representations of the task prompt and the image / video. In the case of the base lines, these representations are fixed (eg. CLIP / Intern2Video representations), whereas for the proposed method these are taken from the last layer of the model learnt on the data itself. This is also reflected in the compute budget - the model takes 5 days to train (in addition to the 5 hours of training in imagination).

**Questions:**

- What is the value of $k$ (number of frames predicted by the connector)? What duration does this correspond to? What happens if the task is longer than this duration?
-  Just to confirm, in the offline RL evaluation, first the world model is trained on the offline data (only for that particular domain) and then the policy is trained in imagination? In that case, why is there a difference in the time taken for the actor-critic to converge in the data-free setting (see line 524)
-  In the single task, offline RL case, is the aligner trained with only as many language prompts as the tasks in that domain? If that is the case, it would be trained to reconstruct $e^{(v)}$ corresponding to many different videos in the offline dataset, some of which might contain suboptimal trajectories which have nothing to do with the language prompt. How can we expect the aligner to learn anything useful in this case?
- If the aligner is trained only on a few language prompts, how is it able to generalize to new tasks?
- What exactly is the multi-task generalization setting? In this evaluation, does the method get access to offline data from the OOD task? If yes, how is it used to train the policy? If no, how are the model-free baselines trained in this setting?

**Limitations:**

Yes, the authors adequately assessed the limitations.

---

> ### Author Rebuttal · Authors · 2024-08-06
>
> We thank the reviewer for the helpful comments.
>
> **Connector-aligner generalization**
>
> First, we would like to make clear that, as stated in multiple parts of the paper, the connector and aligner networks are trained using **vision-only data** and **no language annotations**. We have provided additional clarification for this in our main rebuttal message.
>
> The connector-aligner synergy in the MFWM allows to rely on the multimodal foundation model's knowledge for grounding visual and language prompts into the embodied domains dynamics. The extent to which this system generalizes depends on the knowledge possessed by the two main components: (i) the foundation VLM and (ii) the world model. Given a certain concept, expressed through a prompt, GenRL is able to understand the concept and learn the corresponding behavior given two conditions:
>
> * the VLM has been pre-trained on vision-language pairs that allow understanding that concept. We assume this is the case as foundation models are trained on massive datasets;
> * the world model has been trained on a dataset that contains visual observations that match the given concept. Given a mixed dataset, containing many tasks and exploration data, we know this is true for the tasks in the dataset, but we cannot know for other tasks beforehand.
>
> In practice, we show that the tasks we deliberately put in the dataset can be retrieved (Behavior retrieval experiments) and the system also generalizes to many other prompts/tasks. This means the behaviors are likely to be found in the exploratory data or as part of the tasks data. We assess this generalization capability quantitatively, in the Multitask generalization experiments, and qualitatively, with the visualizations on the website.
>
> **Stronger baselines**
>
> As we describe in the main rebuttal message, we added new model-based baselines to our main experiments. These baselines follow the recommendation of Reviewer Yb6H of learning an "inverse connector" from the world model representation to the VLM representation (GenRL does the opposite). This baseline is stronger than the model-free baselines and it allows as to more clearly establish the main ingredients that contribute to the stronger performance of GenRL: (i) using a video-language model helps in dynamic tasks, (ii) using a model-based algorithm is beneficial, (iii) the connection-alignment system presented outperforms the other straightforward way of connecting the two representations (world model and VLM).
>
> As for the data-free settings, we do not have knowledge of any other data-free behavior learning strategies. As, to the best of our knowledge, this is a new paradigm for behavior learning, we aim to establish its performance compared to more established paradigms, e.g. offline RL. We have provided additional information about our unique data-free pipeline in the main rebuttal message.
>
> **Missing implementation details**
>
> The following information has been added to the paper:
>
> **Value of k**: we adopt a number of frames $k=8$ (as stated at Line 158. We also added this information at Line 104 to make it clearer)
>
> **Multi-task generalization settings**: all methods are trained on the same dataset, made of structured data (the behavior retrieval tasks) and unstructured data (exploration data). Assuming that the data distribution is varied enough, many behaviors that are not part of the behavior retrieval tasks are likely observed. The goal of all methods is to understand the received prompt and learn the "best-matching behavior". This can be done from the given dataset, for the model-free RL baselines, or in imagination after pre-training the world model, for the model-based baselines and GenRL. We observed that GenRL is able to succeed in many tasks that we didn't deliberately add to the dataset.
>
> **Training time**
>
> GenRL pretrains a single world model per dynamic, for 5 days, and uses it for learning behaviors in imagination, for 5 hours per task. Model-free RL methods require 7 hours but have no pre-training stage. On a single GPU, model-free RL is faster to train for a small number of runs. GenRL starts becoming advantageous when using the world model for training for more than 60 runs (which is often the case, considering the number of runs = N seeds x M tasks per domain).
>
> When adopting the data-free learning strategy, GenRL doesn't rely on the dataset at all (see main rebuttal message). This halves the time required for training, as there are no data transfers between the CPU (where the dataset is loaded) and the GPU for training.
>
> We hope to have satisfied all the reviewer's concerns and we look forward to receiving updated feedback.

---

> > ### Comment · Reviewer_X1kr · 2024-08-13
> >
> > Thank you for the clarifications, I have updated my score.

---

> > > ### Author Response · Authors · 2024-08-14
> > >
> > > We would like to thank the reviewer for their valuable feedback, which has significantly improved the quality of our work. The additional experiments and clarifications we incorporated based on the reviewer's recommendations have strengthened our presentation.
> > >
> > > We are also pleased to see that our rebuttal positively influenced the reviewer's opinion of our work, leaning towards an acceptance score.

---

### Author Rebuttal · Authors · 2024-08-06

## Training with no language annotations

We stated several times that the system is trained with vision-only data (Fig. 1, Line 46, Line 469) and no language annotations (Line 11, Line 42). Nonetheless, some reviewers expressed doubts on this matter. We believe the source of confusion is the statement at Line 176-177 ("We have removed the explicit reward information about the task and replaced it with a short task description, in language form.").

To improve clarity, we replaced it with:
"The datasets contain no reward information and no text annotations of the trajectories. The rewards for training for a given task must be inferred by the agent, i.e. using the cosine similarity between observations and the given prompt or, in the case of GenRL, using our reward formulation (Eq. 3)."

**How can the system work with language prompts, if it's not trained on language data?**

The connector learns to map visual embeddings from the pre-trained VLM to latent states of the world model. When learning the connector from visual embeddings $e^{(v)}$, we assume it can generalize to the corresponding language embedding $e^{(l)}$ if the angle $\theta$ between the two embeddings is small enough, see Fig. 11a in attached PDF. This can be expressed as $\cos{\theta} > c$ or $\theta < \arccos{c}$, with $c$ a small positive value [1].

Previous work [1,2] leverages noise during training (of the connector), leading to the situation in Fig. 11b, where $c$ grows larger with the noise. This allows language embeddings to be close enough to their visual counterparts.

In our work, we instead learn an aligner network, which maps points surrounding $e^{(v)}$ closer to $e^{(v)}$ (Fig. 11c). This way $c$ is unaltered but the aligner will map $e^{(l)}$ close enough to $e^{(v)}$. Since we use noise to sample points around $e^{(v)}$ the model can be trained using vision-only data (no language annotations).

We hope this provides a cleaner explanation, which should replace the previous one (Line 118-127).

[1] LAFITE: Towards Language-Free Training for Text-to-Image Generation, Zhou et al

[2] Connect, Collapse, Corrupt: Learning Cross-Modal Tasks with Uni-Modal Data, Zhang et al

## Data-free settings

After pre-training the MFWM, we claim that is possible to train policies for new tasks in a data-free fashion. **How does this differ from the standard GenRL setting and offline zero-shot RL methods?** We answer this question in Fig. 12 (attached PDF).

Offline RL methods (Fig. 12a), combined with VLMs, can learn to perform tasks zero-shot from new prompts, but they need to sample observations and actions from the dataset for computing rewards and for policy learning.

GenRL (Fig. 12b), and potentially other model-based RL methods combined with VLMs, need to sample (sequences of) observations from the dataset, to infer the initial latent states for learning in imagination. Afterwards, rewards can be computed on the imagined latent sequences, enabling policy learning.

In addition, for data-free GenRL (Fig. 12c), we sample the initial latent states internally by combining: (i) random samples of the latent space, (ii) randomly sampled embeddings, which are mapped to "actual embeddings" using the aligner, and turned into latent states, by the connector. Thus, policy learning requires no data sampling at all.

Finally, following the suggestion of Rev. Yb6H, we renamed this paradigm "Data-free policy learning". To the best of our knowledge, there are no previous works that can learn multiple policies in a data-free fashion (after pre-training of a model) which is the reason why we are unable to provide additional baselines.

## Baselines

We added model-based baselines to our main experiments. These baselines follow the recommendation of Rev. Yb6H of learning an "inverse connector" from the world model representation to the VLM representation (GenRL does the opposite). The "inverse connector", given the latent state corresponding to a certain observation, predicts the corresponding embedding. Formally:

Inverse connector: $\hat{e}^{(v)}_t = f(s_t, h_t) $

$\mathcal{L_{inv-conn}} =  || e^{(v)} - \hat{e}^{(v)}_t ||^2_2$

After training the inverse connector, visual embeddings can be inferred from latent states. For policy learning, rewards are computed using the cosine similarity between embeddings inferred from imagined latent states and the prompts' embedding. We call this method **WM-CLIP**.

The inverse connector is implemented as a 4-layer MLP, with a hidden size of 1024. For a fair comparison, we adopt the same world model for WM-CLIP and GenRL. For WM-CLIP we pre-train the additional inverse connector, while for GenRL the connector and aligner. We use one world model for each domain and then train N policies (for N seeds). We have also re-run the main experiments, increasing the number of seeds to 10 for all methods.

In Table 5 and Fig. 13 (attached PDF), we observe that WM-CLIP is stronger than the model-free baselines. This allows us to clearly establish the main ingredients that contribute to the stronger performance of GenRL: (i) the video-language model helps in dynamic tasks, (ii) model-based algorithms lead to higher performance, (iii) the connection-alignment system presented outperforms the "inverse" way of connecting the two representations.

## Behaviors from video prompts

In Fig. 14 of the attached PDF, we provide behavior learning results from video prompts. The videos are also on the project website. The tasks included are static and dynamic, across 4 different domains.

The results show a similar trend to the language prompts experiments and the performance of using video prompts is aligned to language prompts, for the same tasks. In general, we found it interesting that the VLM allows us to generalize to very different visual styles (drawings, realistic, AI-generated), very different camera viewpoints (quadruped, microwave), and different morphologies (cheetah tasks).

---

### Author Response · Authors · 2024-08-06

We would like to thank all the reviewers for their extensive feedback on our submission.

We were particularly glad to read positive comments from the reviewers, who found our results good and surprising (Rev. X1kr), the idea of the paper nice (Rev. jfoT and Yb6H), and the problem interesting and significant (all reviewers).

Reviewers have expressed concerns regarding some aspects of the work, mainly: request for additional baselines (Rev. X1kr, jfoT, Yb6H), additional insights and results learning from video prompts (Rev. Yb6H, jfoT), request for additional details on the experimental settings (Rev. X1kr, jfoT, Yb6H).

In addition, there seems to be a lack of clarity about two main aspects of the paper: (i) how do we train and generalize in the absence of language annotations in the dataset, (ii) how does data-free learning differ from other paradigms, e.g. zero-shot offline RL.

With our rebuttal, we aim to provide clarification over these matters and to provide the reviewers with the additional experiment, information and materials they requested.

Concerning some missing/unclear experimental details, we have replied to each reviewer in their rebuttal section. In addition, we are providing the Area Chair with a link to an anonymized code repository, in case the reviewers would like to verify any additional implementation details.

---

### Decision · Program_Chairs · 2024-09-25

**Decision:**

Accept (poster)

**Comment:**

This work explores utilizing large-scale pre-trained models for interactive & embodied tasks via multimodal world models that generalize across domains. This is achieved by training connector and aligner networks, using vision-only data, to align the latent space of large-scale pre-trained models with that of the generative world models.

In initial reviews, while the submission received mixed overall ratings, all reviewers praised the novelty of the proposed approach and the valuable contribution of leveraging large-scale pre-trained models for RL and embodied AI. During the discussion phase, two of the reviewers (X1kr, Yb6H) moved from borderline rejects to borderline accepts, based on clarifications provided by the authors regarding language annotations, the role of connector & aligner networks, introduction of new baselines and ablations, and authors’ commitment to improving the clarity of the draft. Another reviewer (bBQY) implicitly championed the submission by raising their rating further to a clear acceptance. Part of the committee expressed valid concerns regarding overstatements of the contributions and some confusion about key components of the proposed methodology. The authors engaged in constructive discussions with the reviewers to ground their claims more concretely, clarify technical details, and improve the overall explanation. Reviewer jfot found the new edits warranted a fresh round of reviews. However, they recognized the core idea of the submission as innovative, the results as suitable and indicative of generalization, and the positioning and scope of the work as relevant to the community.

The AC recommends acceptance and agrees with the largely shared opinion of the committee. This work introduces a novel, pertinent, and timely contribution -- that has the potential to significantly benefit the community. The authors are strongly encouraged to improve the draft with the discussed writing edits, new experimental analysis, and provide a clearer description of the key components of their methodology in the final draft.